# On the hypercomplex numbers and normed division algebras in all dimensions: A unified multiplication

**Pushpendra Singh** [1]*, **Anubha Gupta**[2], **Shiv Dutt Joshi**[3]

**1** School of Engineering, Jawaharlal Nehru University, Delhi, India, **2** SBILab, Department of ECE, IIIT-Delhi, Delhi, India, **3** Department of Electrical Engineering, IIT Delhi, Delhi, India

\* spushp@jnu.ac.in

**Data Availability Statement:** The MATLAB codes for the proposed hypercomplex numbers and normed division algebras in all finite dimensions have been made publicly accessible for download, verification of the various properties, and

## Abstract

Mathematics is the foundational discipline for all sciences, engineering, and technology, and the pursuit of normed division algebras in all finite dimensions represents a paramount mathematical objective. In the quest for a real three-dimensional, normed, associative division algebra, Hamilton discovered quaternions, constituting a non-commutative division algebra of quadruples. Subsequent investigations revealed the existence of only four division algebras over reals, each with dimensions 1, 2, 4, and 8. This study transcends such limitations by introducing generalized hypercomplex numbers extending across all dimensions, serving as extensions of traditional complex numbers. The space formed by these numbers constitutes a non-distributive normed division algebra extendable to all finite dimensions. The derivation of these extensions involves the definitions of two new $\pi$-periodic functions and a unified multiplication operation, designated as spherical multiplication, that is fully compatible with the existing multiplication structures. Importantly, these new hypercomplex numbers and their associated algebras are compatible with the existing real and complex number systems, ensuring continuity across dimensionalities. Most importantly, like the addition operation, the proposed multiplication in all dimensions forms an Abelian group while simultaneously preserving the norm. In summary, this study presents a comprehensive generalization of complex numbers and the Euler identity in higher dimensions, shedding light on the geometric properties of vectors within these extended spaces. Finally, we elucidate the practical applications of the proposed methodology as a viable alternative for expressing a quantum state through the multiplication of specified quantum states, thereby offering a potential complement to the established superposition paradigm. Additionally, we explore its utility in point cloud image processing.

## 1 Introduction

The set of real numbers ($\mathbb{R}$) forms a completely ordered field, where addition, subtraction, multiplication, and division are well defined. Imaginary numbers emerged from the mathematical pursuit of finding solutions to quadratic and cubic equations, mainly when these

reproduction of the results and figures in the simulation section. The repository is available on GitHub via the following link: \url{https://github.com/spushp/MATLAB_HypercomplexNumbers}.

**Funding:** The author(s) received no specific funding for this work.

**Competing interests:** The authors have declared that no competing interests exist.

equations presented challenges that could not be resolved within the framework of real numbers. The amalgamation of real and imaginary numbers forms complex numbers ($\mathbb{C}$) that are algebraically closed but not ordered. While $\mathbb{R}$ is a vector space of dimension one over itself, $\mathbb{C}$ is a vector space of dimension two over $\mathbb{R}$. Specifically, $\mathbb{C}$ allows for operations with complex numbers of the form ($a + ib$) and visualizations in 2-dimensional (2D) space, incorporating vector length, distance, and angles. This connection inspired William Rowan Hamilton to seek a 3D algebra with an associated 3D geometry, aiming for a 3D normed division algebra. In October 1843, Hamilton [1] discovered quaternions ($\mathbb{H}$), and famously carved their fundamental equations into the stone of the Brougham Bridge as $i^2 = j^2 = k^2 = ijk = -1$. Quaternions are non-commutative because $ij = -ji$. Octonions ($\mathbb{O}$), non-commutative and non-associative division algebras, were introduced by John Graves in 1843, a contemporary and associate of Hamilton [2]. Arthur Cayley independently discovered octonions, highlighting their mathematical significance [3]. The Cayley–Dickson construction, pioneered by Cayley, systematically generates a $2n$-dimensional algebra from an $n$-dimensional algebra over the real numbers ($\mathbb{R}$) [4]. Applied to octonions, this construction derives an eight-dimensional algebra ($\mathbb{O}$) from quaternions ($\mathbb{H}$), extending complex numbers ($\mathbb{C}$). The unique non-commutative and non-associative properties of octonions make them a significant subject in various mathematical domains, including algebraic theory and theoretical physics. The foundational contributions of Graves and Cayley, along with the efficacy of the Cayley–Dickson construction, underscore the importance of octonions in a broader mathematical discourse.

It is now established that a 3D normed division algebra does not exist. In 1878, Frobenius [5] classified associative normed division algebras, proving that the only three such algebras are $\mathbb{R}$, $\mathbb{C}$, and $\mathbb{H}$. In 1898, Hurwitz [6] extended this to four normed division algebras: $\mathbb{R}$, $\mathbb{C}$, $\mathbb{H}$, and $\mathbb{O}$, with a natural embedding $\mathbb{R} \subset \mathbb{C} \subset \mathbb{H} \subset \mathbb{O}$, where multiplication by a unit vector is distance-preserving. Zorn in 1930 [7] showed that relaxing associativity to alternativity still results in only four normed division algebras: $\mathbb{R}$, $\mathbb{C}$, $\mathbb{H}$, and $\mathbb{O}$. Theorems by Adams (1958, 1960) [8, 9], Kervaire (1958) [10], and Bott–Milnor (1958) [11] confirm that finite-dimensional normed division algebras can only have 1, 2, 4, or 8 dimensions. Recently, a non-distributive three-dimensional real scator algebra was proposed [12], introducing necessary conditions to avoid zero divisors [12].

The work of Hamilton on quaternions was seminal, finding applications in astronautics, robotics, computer graphics, and animation. Quaternions are useful in modern physics, particularly in general relativity, as they can express the Lorentz transform [13]. The quaternion calculus is useful in crystallography, kinematics of rigid body motion, classical electromagnetism, and quantum mechanics [13]. Hamilton sought a real, normed, three-dimensional, associative, division algebra that did not exist. To equate the Euclidean length of the product of triples to the product of their lengths, he dropped commutative multiplication and added a fourth dimension defined by $k$, thus defining a 4D hypercomplex number system instead of a 3D one. Octonions, the 8D hypercomplex number system, drop both commutativity and associativity. For example, in quaternions (4D), a polynomial of degree $n$ can have infinitely many roots, unlike the fundamental theorem of algebra for the 2D complex number system, which guarantees $n$ complex roots (counting multiplicity). Only four real division algebras with dimensions 1, 2, 4, and 8 can exist, as the framework cannot be extended to other finite dimensions [6–11, 14].

Intrigued by these limitations and inspired by Hamilton's unconventional solution, this study proposes an alternative applicable across all dimensions. The distributivity property of the field is omitted, following methodologies in quaternion (non-commutative) and octonion (non-commutative and non-associative) studies. A novel multiplication operation and two distinct $\pi$-periodic functions are introduced, forming a non-distributive normed division

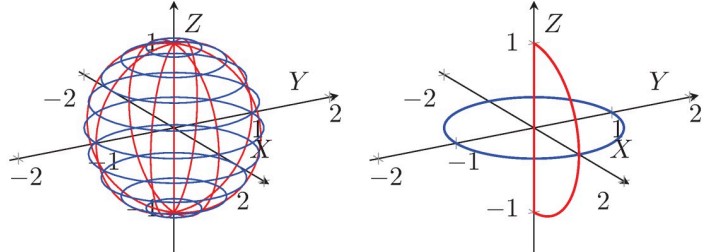

**Fig 1. The proposed multiplication represents a sphere ($re^{i\phi}e^{j\theta}$) as the multiplication of the unit circle ($e^{i\phi}$) in the *XY* plane with the unit semicircle ($e^{j\theta}$) in the *XZ* plane and scaling by radius *r*, as shown in (24) and (27). Here we have considered fixed radius *r*, variable angles $-\pi \leq \phi < \pi$, and $-\pi/2 \leq \theta \leq \pi/2$ to obtain sphere.**

algebra. Despite its non-distributive nature, the algebra exhibits commutative and associative properties across all dimensions. This new algebra is intriguing and generalizable to all finite higher dimensions. This work contributes significantly in the following respects:

1. Introduces generalized hypercomplex numbers of all dimensions ($\mathbb{S}^M$), extending traditional complex numbers with natural nesting as $\mathbb{S} \subset \mathbb{S}^2 \subset \mathbb{S}^3 \cdots \subset \mathbb{S}^{M-1} \subset \mathbb{S}^M$, where $\mathbb{S} = \mathbb{R}$, $\mathbb{S}^2 = \mathbb{C}$, and $M \in \mathbb{Z}^+$.

2. The defined set of hypercomplex numbers $\mathbb{S}^M$ forms a *non-distributive normed division algebra* to all finite higher dimensions for $M \geq 3$ and distributive for $M = 1, 2$.

3. To maintain consistency with the traditional theory of $\mathbb{R}$ and $\mathbb{C}$ spaces, a new multiplication operation called spherical multiplication (SM) is introduced. Unlike traditional multiplication, SM does not follow the distributive property, leading to non-distributive normed division algebra.

4. The proposed SM geometrically represents a sphere as the multiplication of the unit circle in the *XY* plane with the unit semicircle in the *XZ* plane and scaling by radius (e.g., refer Fig 1).

5. The proposed structure extends Euler's identity to all dimensions.

6. Unlike quaternions and octonions, these generalized hypercomplex number systems have finite number of roots for polynomials of degree *n* (e.g., see Example 4).

7. These hypercomplex numbers and corresponding algebras reduce to distributive normed algebras for dimensions 1 and 2, showing backward compatibility and proper generalization of $\mathbb{C}$ in higher dimensions.

8. The proposed multiplication operation in all dimensions forms an Abelian group and preserves the norm, designated as normed Abelian group (NAG).

9. Introduces a generalized vector space equipped with vector addition, scalar multiplication and vector multiplication in all dimensions, thus forming a non-distributive vector algebra.

10. The proposed spherical multiplication provides an invertible nonlinear map.

11. Provides an alternative for expressing a quantum state as the multiplication of specified quantum states, complementing the superposition paradigm in quantum mechanics and computing.

Classical theorems have long restricted normed division algebras to real numbers, complex numbers, quaternions, and octonions. This work transcends these limitations by introducing unified scaling and rotative multiplication, relaxing distributivity, and defining two novel $\pi$-periodic functions, $cos\pi(\theta)$ in (4) and $sin\pi(\theta)$ in (5), as illustrated in Fig 2. These functions enable constructing normed division algebras beyond the classical four dimensions, opening new algebraic research frontiers.

This work transforms hypercomplex number theory by transcending the four-dimensional barrier, constructing real division algebras in all finite dimensions. A unified multiplication scheme facilitates cross-dimensional analysis and extends familiar concepts such as the polar form to higher dimensions. However, the elegance of non-distributive multiplication reveals novel challenges and research avenues. The quest for normed division algebras across all dimensions has driven mathematical advancement for centuries. Although conventional structures like real, complex, quaternionic, and octonionic numbers exist, developing a non-distributive normed division algebra in all dimensions represents a significant theoretical breakthrough. This achievement fills a historical gap, potentially opening new avenues in pure mathematics, particularly for those studying algebraic structures. Despite not offering immediate practical applications, such an algebra can find niche solutions in specific mathematical domains like abstract theories, advanced physics, or specialized fields where non-distributivity is beneficial. Grasping its potential would demand expertise in relevant areas. Maintaining commutativity and associativity sets this algebra apart from non-commutative structures like quaternions and octonions, granting it broader theoretical significance. Innovations such as these showcase the dynamic nature of mathematics, where relaxing foundational properties

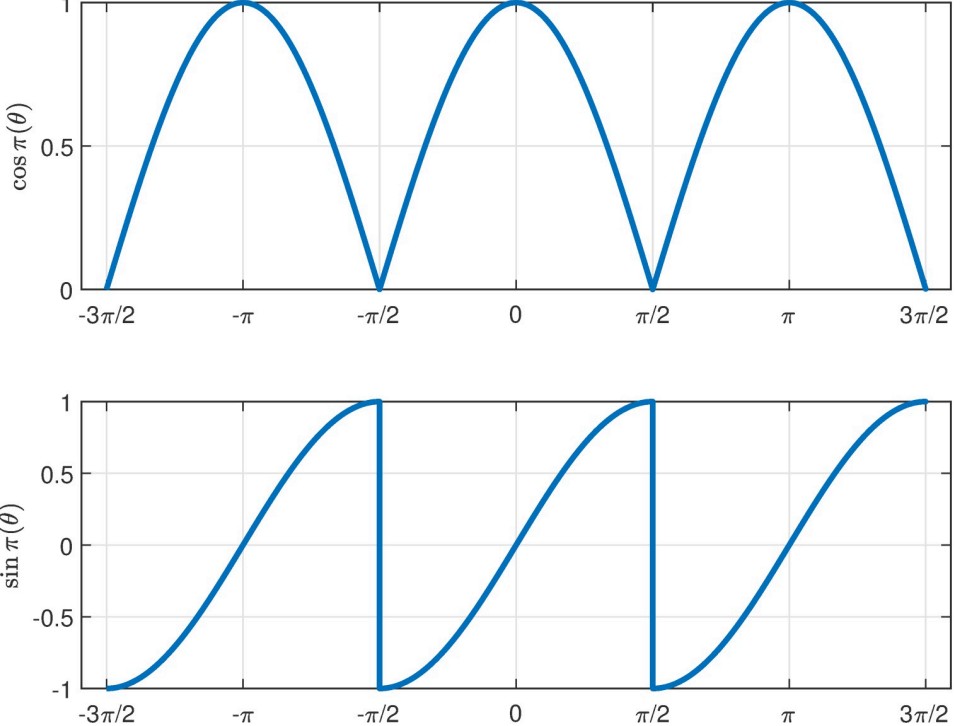

**Fig 2. The $\pi$-periodic functions cos $\pi(\theta)$ (top) and sin $\pi(\theta)$ (bottom) plotted for $[-3\pi/2, 3\pi/2]$.** The function cos $\pi$ ($\theta$) is continuous and sin $\pi(\theta)$ is discontinuous at odd multiples of $\pi/2$, and both functions are differentiable on $\mathbb{R} \setminus \{\frac{\pi}{2} + k\pi \mid k \in \mathbb{Z}\}$. Moreover, within the open interval $(-\pi/2, \pi/2)$ of the principal range, both are continuous and differentiable.

can lead to powerful advancements and diverse applications. The proposed algebraic structure advances complex analysis by generalizing Euler's identity to all dimensions, as presented using scator algebra in [15]. This identity, rooted in complex numbers, reveals a fundamental interconnection among mathematical constants and functions. Extending it to higher dimensions promises broader understanding and potential novel insights and applications across disciplines utilizing multi-dimensional frameworks. This groundbreaking work reshapes our understanding of hypercomplex numbers, leaving an indelible mark on abstract algebra and the mathematical landscape by demonstrating the adaptability of structure and its ability to address challenges and pave the way for future exploration across various disciplines.

The subsequent sections of this work are structured as follows: Section 2 provides the foundational information. Section 3 expounds upon the proposed 3-dimensional (3D) hypercomplex number system with examples. Section 4 introduces the proposed generalized $M$-dimensional ($M$D) hypercomplex number system. A generalized real vector space is presented in Section 5. Spherical multiplication as an invertible nonlinear map is presented in Section 6. Relationship between addition and multiplication through logarithms is presented in Section 7. Section 8 covers numerical simulation results concerning the Bloch sphere representation in quantum mechanics and computing, and its application in point cloud image processing.

## 2 Preliminaries

A field is a triplet $(F, +, \cdot)$ where $F$ is a set, two binary operations on $F$ called addition (+) and multiplication ($\cdot$) where the binary operation on $F$ is a mapping $F \times F \to F$ such that it satisfies the following field axioms for all $g_1, g_2, g_3 \in F$

1. Closure of addition and multiplication: $g_1 + g_2 \in F$ and $g_1 \cdot g_2 \in F$

2. Associativity of addition and multiplication: $g_1 + (g_2 + g_3) = (g_1 + g_2) + g_3$, and $g_1 \cdot (g_2 \cdot g_2) = (g_1 \cdot g_2) \cdot g_3$

3. Commutativity of addition and multiplication: $g_1 + g_2 = g_2 + g_1$ and $g_1 \cdot g_2 = g_2 \cdot g_1$

4. Additive and multiplicative identities: for every $g \in F$, there exist two elements 0 and 1 in $F$ such that $g + 0 = g$ and $g \cdot 1 = g$

5. Additive and multiplicative inverses: for every $g \in F$, $\exists g_{ai}$ or $-g$ in $F$, called the additive inverse of $g$, such that $g + g_{ai} = 0$; and for every nonzero $g \in F$, $\exists g^{-1}$ or $1/g$ in $F$, called the multiplicative inverse of $g$, such that $g \cdot g^{-1} = 1$

6. Distributivity of multiplication over addition:
   $g_1 \cdot (g_2 + g_3) = g_1 \cdot g_2 + g_1 \cdot g_3$.

Some prominent examples of fields include the set of real numbers ($\mathbb{R}$), the set of complex numbers ($\mathbb{C}$), and the set of rational numbers ($\mathbb{Q}$). In algebra, a structure in which multiplication is not commutative is known as a skew field or a division ring. The quaternions are a well-known example of a non-commutative division ring. Furthermore, we define the algebraic structure designated as a *non-distributive field* (NDF) that adheres to all axioms of a field except for the distributivity of multiplication over addition.

The distributive property of multiplication over addition is a natural consequence of the historical development of multiplication as an operation derived from repeated addition. While multiplication was indeed historically derived from repeated addition, the reverse is not true as addition does not distribute over multiplication. The distributive property is a distinctive feature of multiplication and is not a property shared by addition in the same manner. The distributive property is a fundamental tool in algebra that simplifies expressions, aids in

factoring, maintains consistency in operations, facilitates computation, and serves as a building block for further algebraic manipulation and understanding.

# 3 Proposed 3D hypercomplex number system

In this section, first, we define three-dimensional (3D) hypercomplex numbers (denoted as the set $\mathbb{S}^3$) as a true extension of existing two-dimensional complex numbers ($\mathbb{C}$), which we denote as $\mathbb{S}^2$, i.e., $\mathbb{C} = \mathbb{S}^2$.

## 3.1 Proposed 3D hypercomplex number system

We consider a 3D number from the set $\mathbb{S}^3$, in the Cartesian coordinate system (CCS), as

$$g = a + ib + jc, \tag{1}$$

such that the set $\{1, i, j\}$ is basis where $i$ and $j$ are two imaginary numbers, and $a, b, c \in \mathbb{R}$. This 3D number (1) can also be written using the triplet notation as: $g = \begin{bmatrix} a \\ b \\ c \end{bmatrix}$. First, we write (1) in one of the standard spherical coordinate system (SCS) as

$$
\begin{aligned}
a &= r\cos(\theta)\cos(\phi), \quad b = r\cos(\theta)\sin(\phi), \quad c = r\sin(\theta), \\
r &= \sqrt{a^2 + b^2 + c^2}, \quad \phi = \arctan\left(\frac{b}{a}\right), \quad \theta = \arctan\left(\frac{c}{\sqrt{a^2 + b^2}}\right), \\
g(r, \phi, \theta) &= r\cos(\theta)\cos(\phi) + ir\cos(\theta)\sin(\phi) + jr\sin(\theta),
\end{aligned}
\tag{2}
$$

where $r \in [0, \infty)$, $\phi \in [0, 2\pi)$ and $\theta \in [-\pi/2, \pi/2]$. The function atan2($b, a$) computes the four-quadrant inverse tangent arctan($b/a$) and utilizes the signs and magnitudes of $b$ and $a$ to determine the correct quadrant for $\phi$. Moreover, we can add $2\pi n$ for $n \in \mathbb{Z}$ to both angles $\phi$ and $\theta$ without changing the point, i.e., $g(r, \phi, \theta) = g(r, \phi + 2\pi n, \theta + 2\pi n)$. Thus, a spherical coordinate triplet $(r, \phi, \theta)$ specifies a single point in a three-dimensional space that has infinitely many equivalent spherical coordinates.

**Remark 1**. We can fix $n = 0$ to remove much of the non-uniqueness in the representation of a point in the SCS. In spite of everything, persistent nonuniqueness in representations arises in two specific scenarios within the SCS:

1. Origin ($r = 0$): In this case, when the radial coordinate $r$ assumes a value of zero, the angular coordinates $\phi$ and $\theta$ become unrestricted, resulting in an infinite representation of the origin as $(0, \phi, \theta)$ in the SCS. This signifies that any non-zero vector $(r, \phi, \theta)$ can be asymptotically approached to a zero vector by allowing its radial component $r$ to approach zero, akin to the behavior observed in polar representations of two-dimensional complex numbers. In other words, variations in the angular coordinates $\phi$ or $\theta$ while situated at the origin do not effectuate any displacement of the vector.

2. Z-axis ($\theta = \pm\pi/2$): In this specific scenario where $\theta$ equals $\pi/2$ or $-\pi/2$, for a fixed $r$, the azimuth angle $\phi$ can assume any value, as explicitly revealed by equation (2). Consequently, a vector situated along the Z-axis possesses an infinite array of representations as $(r, \phi, \pm\pi/2)$. This implies that vectors characterized by diverse values of $r$, $\phi$, and $\theta$ can be aligned along the Z-axis by adjusting $\theta$ to $\pm\pi/2$. Once aligned along the Z-axis, alterations in the polar angle $\phi$ fail to induce any alteration in the orientation of the vector. Thus, in these instances, manipulation of the values of one or more of the other coordinates can be undertaken without inducing displacement of the point in question.

Subsequently, in pursuit of the specific objectives delineated in Remark 2 and aiming to construct new multiplication (16) and division (17) operations with demonstrably sound properties, we propose a meticulously refined definition for (2) as follows:

$$a = r \cos\!\pi(\theta)\cos(\phi), \quad b = r \cos\!\pi(\theta)\sin(\phi), \quad c = r \sin\!\pi(\theta),$$

$$r = \sqrt{a^2 + b^2 + c^2}, \quad \phi = \arctan\left(\frac{b}{a}\right) \in [0, 2\pi), \quad \theta = \arctan\left(\frac{c}{\sqrt{a^2 + b^2}}\right) \in [-\pi/2, \pi/2], \quad (3)$$

where the $\pi$-periodic functions, as shown in Fig 2, are defined for all $\theta \in \mathbb{R}$ as

$$\cos\!\pi(\theta) = \begin{cases} \cos(\theta), & \text{if } \theta \in \bigcup_{n=-\infty}^{\infty}\left[-\frac{\pi}{2} + 2\pi n, \frac{\pi}{2} + 2\pi n\right], \\[2mm] -\cos(\theta), & \text{if } \theta \in \bigcup_{n=-\infty}^{\infty}\left(\frac{\pi}{2} + 2\pi n, \frac{3\pi}{2} + 2\pi n\right), \end{cases} \quad (4)$$

$$\text{and} \quad \sin\!\pi(\theta) = \begin{cases} \sin(\theta), & \text{if } \theta \in \bigcup_{n=-\infty}^{\infty}\left[-\frac{\pi}{2} + 2\pi n, \frac{\pi}{2} + 2\pi n\right], \\[2mm] -\sin(\theta), & \text{if } \theta \in \bigcup_{n=-\infty}^{\infty}\left(\frac{\pi}{2} + 2\pi n, \frac{3\pi}{2} + 2\pi n\right), \end{cases} \quad (5)$$

where the intervals form a complete, non-overlapping partition of the real line as

$$\mathbb{R} = \left\{ \bigcup_{n=-\infty}^{\infty}\left[-\frac{\pi}{2} + 2n\pi, \frac{\pi}{2} + 2n\pi\right] \right\} \bigcup \left\{ \bigcup_{n=-\infty}^{\infty}\left(\frac{\pi}{2} + 2n\pi, \frac{3\pi}{2} + 2n\pi\right) \right\}. \quad (6)$$

We observe that $\cos\!\pi(\theta) = \cos\!\pi(\theta + n\pi) = |\cos(\theta)|$ for all $n \in \mathbb{Z}$ and $\theta \in \mathbb{R}$. Moreover, $\sin\!\pi(\theta) = \sin\!\pi(\theta + \pi)$ for $\theta \neq -\pi/2$ and $\sin\!\pi(\theta) = -\sin\!\pi(\theta + \pi)$ for $\theta = -\pi/2$. Therefore, $\sin\!\pi(\theta) = \sin\!\pi(\theta + n\pi)$ for all $n \in \mathbb{Z}$ and for almost all $\theta \in \mathbb{R}$. Thus, $\sin\!\pi(\theta)/\cos\!\pi(\theta) = \tan\!\pi(\theta) = \tan(\theta)$, which is differentiable in $\mathbb{R} \setminus \left\{\frac{\pi}{2} + k\pi \mid k \in \mathbb{Z}\right\}$. The function $\cos\!\pi(\theta)$ is continuous, while $\sin\!\pi(\theta)$ is discontinuous at odd multiples of $\frac{\pi}{2}$, and both functions are differentiable in $\mathbb{R} \setminus \left\{\frac{\pi}{2} + k\pi \mid k \in \mathbb{Z}\right\}$. Moreover, within the open interval $(-\pi/2, \pi/2)$ of the principal range, both $\cos\!\pi(\theta)$ and $\sin\!\pi(\theta)$ are continuous and differentiable. Thus, using these functions, we redefine the SCS representation (2) as

$$g = r \cos\!\pi(\theta)\cos(\phi) + ir \cos\!\pi(\theta)\sin(\phi) + jr \sin\!\pi(\theta), \quad (7)$$

where $g$ is $g(r, \phi, \theta)$, $\phi$ is an azimuth angle, and $\theta$ is an elevation angle form $XY$ plane as shown in Fig 3. The significance of the periodicity of the hypercomplex number $g$, as defined in (7), should be emphasized appropriately. Specifically, it is imperative to recognize that $g(r, \phi, \theta)$ exhibits periodic behavior, as expressed by the relationship $g(r, \phi, \theta) = g(r, \phi + 2m\pi, \theta + m\pi)$ for all integers $m \in \mathbb{Z}$. This mathematical characterization underscores the recurrence of the values with respect to variations in the angular coordinates $\phi$ and $\theta$ and elucidates the systematic nature of $g$ within the specified parameter space. Such periodicity holds broader implications for the analytical treatment and interpretation of the hypercomplex number, serving as a fundamental consideration in the study of its properties and behavior. This is to note that the conventional notation of spherical coordinate system is not considered in this work (One can also use the conventional notation of spherical coordinate system where angle $\theta$ is measured from the $Z$-axis. This will also lead to a 3D hypercomplex number system. For more details,

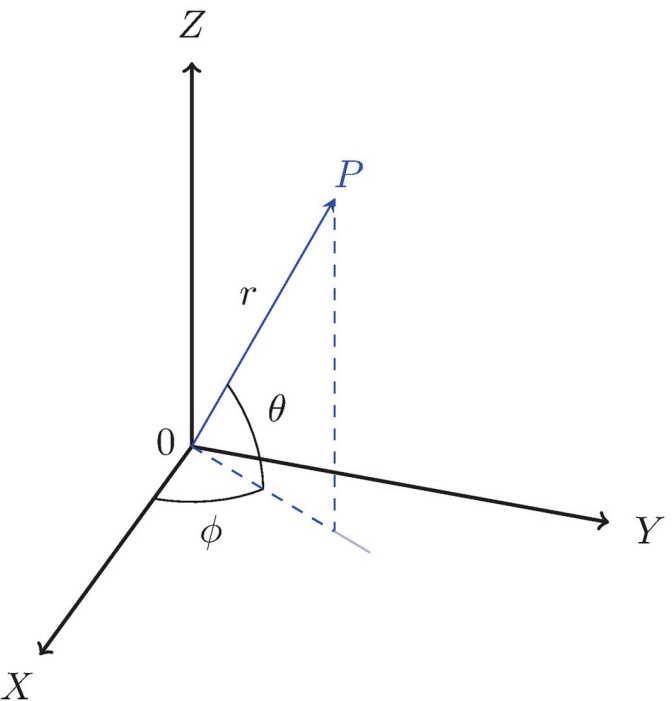

**Fig 3. A point *P* in the considered spherical co-ordinate system, where radius (*r*), azimuth angle (*ϕ* rad), and elevation angle (*θ* rad) measured from *XY* plane are shown.**

see Appendix A.). Thus, from (1), (3) and (7), we can write

$$
g = \begin{bmatrix} a \\ b \\ c \end{bmatrix} = \begin{bmatrix} r\cos\pi(\theta)\cos(\phi) \\ r\cos\pi(\theta)\sin(\phi) \\ r\sin\pi(\theta) \end{bmatrix} \simeq \begin{bmatrix} r \\ \phi \\ \theta \end{bmatrix}, \tag{8}
$$

which represents the following eight cases depending on the three parameters $r$, $\phi$ and $\theta$:

1. A vector that represents a point on a unit sphere for $r = 1$, $\phi = \phi_0$ and $\theta = \theta_0$.

2. A circle for $r = 1$ and $\phi \in [0, 2\pi)$ at elevation angle $\theta = \theta_0$ (e.g., see Fig 1).

3. A semicircle for $r = 1$ and $\theta \in [-\pi/2, \pi/2]$ at azimuth angle $\phi = \phi_0$ (e.g., see Fig 1).

4. A unit sphere for $r = 1$, $\phi \in [0, 2\pi)$ and $\theta \in [-\pi/2, \pi/2]$.

5. A solid unit sphere for $0 \leq r \leq 1$, $\phi \in [0, 2\pi)$ and $\theta \in [-\pi/2, \pi/2]$.

6. A conical surface for $0 \leq r \leq 1$ and $\phi \in [0, 2\pi)$ at elevation $\theta = \theta_0$.

7. A semidisk for $0 \leq r \leq 1$ and $\theta \in [-\pi/2, \pi/2]$ at azimuth $\phi = \phi_0$.

8. A ray for $0 \leq r \leq 1$, $\phi = \phi_0$ and $\theta = \theta_0$.

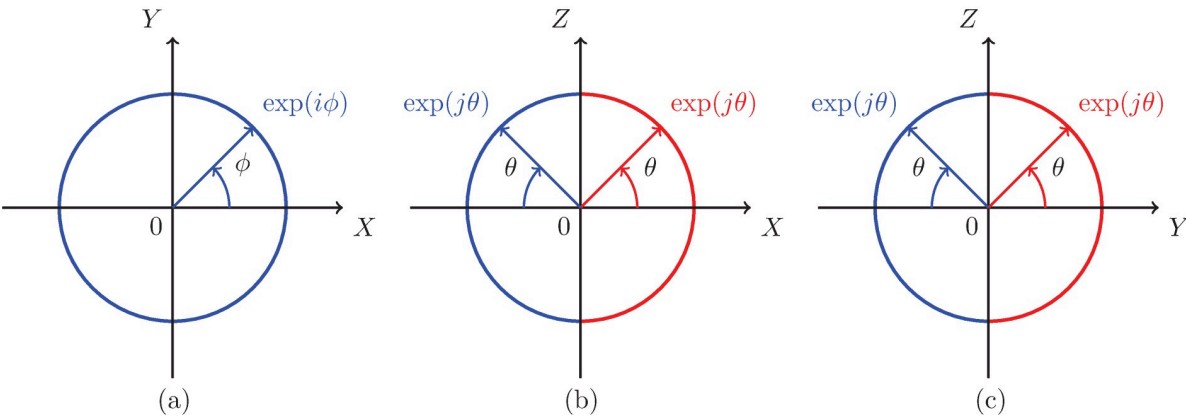

**Fig 4.** Euler identities (a) $\exp(i\phi)$ for $\phi \in [0, 2\pi)$ with $r = 1$ and $\theta = 0$, where a vector that represents a point traces full circle in the direction of arrow as $\phi$ increases, (b) $\exp(j\theta)$ for $\theta \in [-\pi/2, \pi/2]$ with $r = 1$ and $\phi = \pi$ for $X < 0$ (blue semicircle), and $\phi = 0$ for $X > 0$ (red semicircle), where a point moves in the direction of arrow in the right semicircle as $\theta$ increases with $\phi = 0$, and similarly a point moves in the direction of arrow in left semicircle as $\theta$ increases with $\phi = \pi$, (c) $\exp(j\theta)$ for $\theta \in [-\pi/2, \pi/2]$ with $r = 1$ and $\phi = 3\pi/2$ (blue semicircle) and $\phi = \pi/2$ (red semicircle). The path traced by $\exp(j\theta)$ follows a semicircular trajectory, as indicated by the blue and red semicircle in (b) and (c) for the appropriate values of $\phi$.

We use the representation $g = \begin{bmatrix} a \\ b \\ c \end{bmatrix}$ in CCS, and $g = \begin{bmatrix} r \\ \phi \\ \theta \end{bmatrix}$ in SCS, therefore $\begin{bmatrix} a \\ b \\ c \end{bmatrix} \simeq$

$\begin{bmatrix} r \\ \phi \\ \theta \end{bmatrix}$ where this equality is in the sense that both represent the same point. We consider three special cases of (3) as follows:

1. Consider $XY$ plane with $\theta = 0$, as shown in Fig 4(a), which implies

$$
\begin{aligned}
a &= r\cos(\phi), \quad b = r\sin(\phi), \quad c = 0, \\
r &= \sqrt{a^2 + b^2}, \quad \phi = \arctan\left(\frac{b}{a}\right) \in [0, 2\pi)
\end{aligned}
\tag{9}
$$

and thus $\begin{bmatrix} r \\ \phi \\ 0 \end{bmatrix}$ for $r = 1$, $0 \le \phi < 2\pi$ represents the unit circle in $XY$ plane.

2. Consider $XZ$ plane with (i) $\phi = 0$, as shown in Fig 4(b), right semicircle on the $XZ$ plane $\begin{bmatrix} r \\ 0 \\ \theta \end{bmatrix}$ for $r = 1$, $-\pi/2 \le \theta \le \pi/2$, which implies

$$
\begin{aligned}
a &= r\cos\pi(\theta) \ge 0, \quad b = 0, \quad c = r\sin\pi(\theta), \\
r &= \sqrt{a^2 + c^2}, \quad \theta = \arctan\left(\frac{c}{\sqrt{a^2}}\right) \in [-\pi/2, \pi/2],
\end{aligned}
\tag{10}
$$

and (ii) $\phi = \pi$, as shown in Fig 4(b), left semicircle $\begin{bmatrix} r \\ \pi \\ \theta \end{bmatrix}$ for $r = 1$, $-\pi/2 \le \theta \le \pi/2$, which

implies

$$a = -r\cos\pi(\theta) \leq 0, \quad b = 0, \quad c = r\sin\pi(\theta),$$
$$r = \sqrt{a^2 + c^2}, \quad \theta = \arctan\left(\frac{c}{\sqrt{a^2}}\right) \in [-\pi/2, \pi/2]. \tag{11}$$

3. Consider *YZ* plane with (i) $\phi = \pi/2$, as shown in Fig 4(c), right semicircle on the *YZ* plane $\begin{bmatrix} r \\ \pi/2 \\ \theta \end{bmatrix}$ for $r = 1$, $-\pi/2 \leq \theta \leq \pi/2$, which implies

$$a = 0, \quad b = r\cos\pi(\theta), \quad c = r\sin\pi(\theta),$$
$$r = \sqrt{b^2 + c^2}, \quad \theta = \arctan\left(\frac{c}{\sqrt{b^2}}\right) \in [-\pi/2, \pi/2], \tag{12}$$

and (ii) $\phi = 3\pi/2$, as shown in Fig 4(c), left semicircle $\begin{bmatrix} r \\ 3\pi/2 \\ \theta \end{bmatrix}$ for $r = 1$, $-\pi/2 \leq \theta \leq \pi/2$,

which implies

$$a = 0, \quad b = -r\cos\pi(\theta), \quad c = r\sin\pi(\theta),$$
$$r = \sqrt{b^2 + c^2}, \quad \theta = \arctan\left(\frac{c}{\sqrt{b^2}}\right) \in [-\pi/2, \pi/2]. \tag{13}$$

To obtain the generalized addition and multiplication, we write hypercomplex numbers $g, g_1, g_2 \in \mathbb{S}^3$, using SCS representations (2)–(8), in the triplet notations as

$$g = \begin{bmatrix} a \\ b \\ c \end{bmatrix} \simeq \begin{bmatrix} r \\ \phi \\ \theta \end{bmatrix}, \quad g_1 = \begin{bmatrix} a_1 \\ b_1 \\ c_1 \end{bmatrix} \simeq \begin{bmatrix} r_1 \\ \phi_1 \\ \theta_1 \end{bmatrix}, \quad g_2 = \begin{bmatrix} a_2 \\ b_2 \\ c_2 \end{bmatrix} \simeq \begin{bmatrix} r_2 \\ \phi_2 \\ \theta_2 \end{bmatrix}, \quad \text{and} \quad g_3 = \begin{bmatrix} a_3 \\ b_3 \\ c_3 \end{bmatrix} \simeq \begin{bmatrix} r_3 \\ \phi_3 \\ \theta_3 \end{bmatrix}. \tag{14}$$

We consider traditional addition operations as

$$g_1 + g_2 = \begin{bmatrix} a_1 + a_2 \\ b_1 + b_2 \\ c_1 + c_2 \end{bmatrix}. \tag{15}$$

Further, we define a new multiplication operation, named hereby spherical multiplication (SM) as

$$g_1 g_2 = \begin{bmatrix} r_1 r_2 \\ \phi_1 + \phi_2 \\ \theta_1 + \theta_2 \end{bmatrix}. \tag{16}$$

The proposed SM performs scaling and rotating operations. The additive inverse of $g_2$ is

given by $-g_2 = \begin{bmatrix} -a_2 \\ -b_2 \\ -c_2 \end{bmatrix}$, which leads to the definition of the subtraction operation as

$g_1 - g_2 = \begin{bmatrix} a_1 - a_2 \\ b_1 - b_2 \\ c_1 - c_2 \end{bmatrix}$. Similarly, the multiplicative inverse of $g_2$ is defined as $1/g_2 = \begin{bmatrix} 1/r_2 \\ -\phi_2 \\ -\theta_2 \end{bmatrix}$,

allowing us to define the spherical division (SD) operation as

$$g_1/g_2 = \begin{bmatrix} r_1/r_2 \\ \phi_1 - \phi_2 \\ \theta_1 - \theta_2 \end{bmatrix}, \quad \text{where } r_2 \neq 0. \tag{17}$$

The complex conjugate of $g$ presented in (1) is defined as

$$\bar{g} = a - ib - jc \Rightarrow \bar{g} = \begin{bmatrix} a \\ -b \\ -c \end{bmatrix} \simeq \begin{bmatrix} r \\ -\phi \\ -\theta \end{bmatrix}, \tag{18}$$

such that $g\bar{g} = r^2$. The norm of $g$ is defined as $\|g\| = \sqrt{(g\bar{g})} = r$ and satisfies certain properties, such as positivity, homogeneity, and triangle inequality. Thus, the unified multiplication operation defined in (16) consists of scaling and rotation operations such that $\|g_1 g_2\| = \|g_1\|\|g_2\|$, and $\|g_1 + g_2\| \leq \|g_1\| + \|g_2\|$. In contrast, a non-distributive elliptic scator algebra that extends complex numbers to a higher number of dimensions fails to satisfy the triangle inequality of the norm [16].

The defined SM (16) reduces to the traditional multiplication when $g$ moves from 3D to 2D by considering $c = 0$ in (1). The conjugation with respect to $i$ and $j$ can be defined as

$$\bar{g}_i = a - ib + jc \Rightarrow \bar{g}_i = \begin{bmatrix} a \\ -b \\ c \end{bmatrix} \simeq \begin{bmatrix} r \\ -\phi \\ \theta \end{bmatrix} \quad \text{and} \quad \bar{g}_j = a + ib - jc \Rightarrow \bar{g}_j = \begin{bmatrix} a \\ b \\ -c \end{bmatrix} \simeq \begin{bmatrix} r \\ \phi \\ -\theta \end{bmatrix}, (19)$$

respectively. The multiplicative inverse of (1) is defined as $g^{-1} = \frac{\bar{g}}{g\bar{g}} = \frac{a - ib - jc}{a^2 + b^2 + c^2}$ for every $g \neq 0$, which is same as inverse of a quaternion. This can be written in the triplet notation as

$$g^{-1} = \begin{bmatrix} 1/r \\ -\phi \\ -\theta \end{bmatrix}. \tag{20}$$

Further, we can compute the power of $g^\ell$ as

$$g^\ell = \begin{bmatrix} r^\ell \\ \ell\phi \\ \ell\theta \end{bmatrix}, \quad \forall \ell \in \mathbb{Z}, \tag{21}$$

and (21) becomes (20) when $\ell = -1$.

Addition of two complex numbers (e.g., $g_2 + g_3$) can be written as

$$g_2 + g_3 = (a_2 + a_3) + i(b_2 + b_3) + j(c_2 + c_3) \simeq \begin{bmatrix} r_2 \\ \phi_2 \\ \theta_2 \end{bmatrix} + \begin{bmatrix} r_3 \\ \phi_3 \\ \theta_3 \end{bmatrix} = \begin{bmatrix} r_{23} \\ \phi_{23} \\ \theta_{23} \end{bmatrix},$$

and thus

$$g_1(g_2 + g_3) = \begin{bmatrix} r_1 \\ \phi_1 \\ \theta_1 \end{bmatrix} \begin{bmatrix} r_{23} \\ \phi_{23} \\ \theta_{23} \end{bmatrix} = \begin{bmatrix} r_1 r_{23} \\ \phi_1 + \phi_{23} \\ \theta_1 + \theta_{23} \end{bmatrix}. \tag{22}$$

We can also compute $g_1 g_2 + g_1 g_3$ as

$$g_1 g_2 + g_1 g_3 = \begin{bmatrix} r_1 r_2 \\ \phi_1 + \phi_2 \\ \theta_1 + \theta_2 \end{bmatrix} + \begin{bmatrix} r_1 r_3 \\ \phi_1 + \phi_3 \\ \theta_1 + \theta_3 \end{bmatrix} = \begin{bmatrix} r_{123} \\ \phi_{123} \\ \theta_{123} \end{bmatrix}. \tag{23}$$

From (22) and (23), it is easy to verify that, in general, $g_1(g_2 + g_3) \neq g_1 g_2 + g_1 g_3$. Moreover, $g_1(g_2 + g_3) = g_1 g_2 + g_1 g_3$ if $\theta_1 = 0$, i.e., $g_1 \in \mathbb{S}^2$.

Interestingly, similar to $\begin{bmatrix} a \\ b \\ c \end{bmatrix} = \begin{bmatrix} a \\ 0 \\ 0 \end{bmatrix} + \begin{bmatrix} 0 \\ b \\ 0 \end{bmatrix} + \begin{bmatrix} 0 \\ 0 \\ c \end{bmatrix}$, using the proposed multiplication (16), we can write

$$\begin{bmatrix} r \\ \phi \\ \theta \end{bmatrix} = \begin{bmatrix} r \\ 0 \\ 0 \end{bmatrix} \begin{bmatrix} 1 \\ \phi \\ 0 \end{bmatrix} \begin{bmatrix} 1 \\ 0 \\ \theta \end{bmatrix}, \tag{24}$$

where from (7) and (14)

$$\begin{bmatrix} r \\ 0 \\ 0 \end{bmatrix} \simeq r, \quad \begin{bmatrix} 1 \\ \phi \\ 0 \end{bmatrix} \simeq \cos(\phi) + i\sin(\phi) = e^{i\phi}, \quad \begin{bmatrix} 1 \\ 0 \\ \theta \end{bmatrix} \simeq \cos\pi(\theta) + j\sin\pi(\theta) = e^{j\theta}, \tag{25}$$

where $\{e^{i\phi} \mid \phi \in [0, 2\pi)\}$ represents all points on the unit circle in the *XY* plane and $\{e^{j\theta} \mid \theta \in [-\pi/2, \pi/2]\}$ represents all points on the unit semicircle in *XZ* plane. In general, $\phi \in \mathbb{R}$ and $e^{i\phi}$ is $2\pi$-periodic, $\theta \in \mathbb{R}$ and $e^{j\theta}$ is $\pi$-periodic. The $2\pi$ modulo operation is applied to $\phi$ to restrict it to the principal range $[0, 2\pi)$. Similarly, the $\pi$ modulo operation is used to adjust $\theta$ to lie within the principal range $[-\pi/2, \pi/2]$. Specifically, if $\theta \notin [-\pi/2, \pi/2]$, it is normalized to this range by adding or subtracting a multiple of $\pi$. Hence, for the elevation angles, we employ the transformation for all $\theta \in \mathbb{R}$ as follows:

$$\theta \mapsto \theta \,\hat{\mathrm{mod}}\, \pi = \begin{cases} \theta & \text{if } \theta \in [-\pi/2, \pi/2], \\ \theta - m\pi & \text{if } \theta > \frac{\pi}{2}, \\ \theta + m\pi & \text{if } \theta < -\frac{\pi}{2}, \end{cases} \tag{26}$$

where $m$ is the smallest positive integer such that the result lies within the interval $\left[-\frac{\pi}{2}, \frac{\pi}{2}\right]$. Therefore, the azimuth angle $\phi$ is considered $2\pi$-periodic, and the elevation angle $\theta$ is considered $\pi$-periodic as shown in Fig 5. Therefore, $\cos\pi(\theta) = \cos(\theta)$ and $\sin\pi(\theta) = \sin(\theta)$ with $\theta \mapsto \theta \,\hat{\mathrm{mod}}\, \pi \in [-\pi/2, \pi/2]$. Thus (7) can be written as

$$\begin{aligned} r(\cos\pi(\theta)\cos(\phi) + i\cos\pi(\theta)\sin(\phi) + j\sin\pi(\theta)) &= r[\cos(\phi) + i\sin(\phi)][\cos\pi(\theta) + j\sin\pi(\theta)], \\ &= re^{i\phi}e^{j\theta}, \end{aligned} \tag{27}$$

where $[\cos(\phi) + i\sin(\phi)] \times j\sin\pi(\theta) = e^{i\phi} \times j\sin\pi(\theta) = j\sin\pi(\theta)$, $i^2 = -1$ and $j^2 = -1$. Therefore,

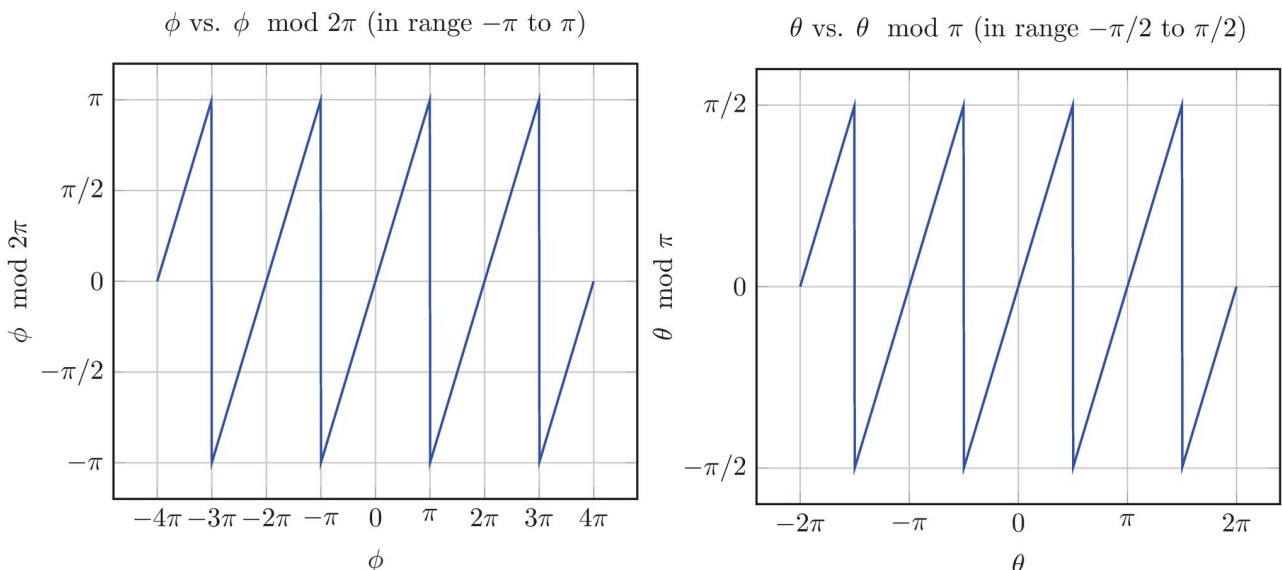

**Fig 5. Plot illustrating the azimuth angle φ versus (φ mod 2π) in the range −π to π, and the elevation angle θ versus (θ mod π) in the range −π/2 to π/2.** The azimuth angle φ is considered 2π-periodic, and the elevation angle θ is considered π-periodic.

geometrically, we have defined a multiplication that represents a sphere as the multiplication of a circle in the *XY* plane with a semicircle in the *XZ* plane. The resulting sphere can be smoothly traversed without encountering any breaks, reflecting the continuous nature of the circle and semicircle components, as shown in Fig 1.

**Remark 2**. To confine the elevation angle $\theta$ within the interval $[-\pi/2, \pi/2]$, prevent the occurrence of dual representations, and ensure consistent multiplication outcomes in accordance with (16), even when the sum $(\theta_1 + \theta_2)$ exceeds the specified range of $[-\pi/2, \pi/2]$, and determine the spatial coordinates of a point in three dimensions post-multiplication, an important modification has been implemented in the SCS (2) for elevation angle. Specifically, the conventional trigonometric functions $\cos(\theta)$ and $\sin(\theta)$ have been replaced with their respective counterparts $\cos\pi(\theta)$ and $\sin\pi(\theta)$, leading to a new representation of SCS (7). This substitution effectively addresses the specified objectives, offering a systematic approach to confining the elevation angle, avoiding dual representations, ensuring consistent multiplication outcomes, and accurately determining the 3D position of a point following the multiplication process.

Using the defined expressions (1), (3) and (14), we can map the CCS basis to SCS as:

$$
1 = \begin{bmatrix} 1 \\ 0 \\ 0 \end{bmatrix} \simeq \begin{bmatrix} 1 \\ 0 \\ 0 \end{bmatrix}; \ i = \begin{bmatrix} 0 \\ 1 \\ 0 \end{bmatrix} \simeq \begin{bmatrix} 1 \\ \pi/2 \\ 0 \end{bmatrix}; \ j = \begin{bmatrix} 0 \\ 0 \\ 1 \end{bmatrix} \simeq \begin{bmatrix} 1 \\ \phi \\ \pi/2 \end{bmatrix}, \tag{28}
$$

where $\phi \in [0, 2\pi)$. The azimuth angle $\phi$ in the expression $j = e^{i\phi}e^{j\frac{\pi}{2}}$ indicates that any subsequent variation in the elevation angle $\theta$ results in the displacement of the point from the *Z*-axis towards the direction specified by $\phi$. To derive the distinctive characterization of the complex unit *j*, subject to the condition $j^2 = -1$, a specific angular value, $\phi = \pi/2$, is established in the context of the identity presented in (28). It is imperative to note that, unless explicitly indicated otherwise, this particular angular assignment shall be consistently adhered to throughout the

course of this investigation. In addition, we can write

$$0 = \begin{bmatrix} 0 \\ 0 \\ 0 \end{bmatrix} \simeq \begin{bmatrix} 0 \\ 0 \\ 0 \end{bmatrix}; \ -1 = \begin{bmatrix} -1 \\ 0 \\ 0 \end{bmatrix} \simeq \begin{bmatrix} 1 \\ \pi \\ 0 \end{bmatrix}; \ 1/i = -i = \begin{bmatrix} 0 \\ -1 \\ 0 \end{bmatrix} \simeq \begin{bmatrix} 1 \\ -\pi/2 \\ 0 \end{bmatrix};$$

$$1/j = -j = \begin{bmatrix} 0 \\ 0 \\ -1 \end{bmatrix} \simeq \begin{bmatrix} 1 \\ -\phi \\ -\pi/2 \end{bmatrix} \tag{29}$$

such that $1(-1) = -1$, $i(-i) = 1$, $j(-j) = 1$, and $0(g) = 0$. The additive identity 0 and the multiplicative identity 1 are distinct and unique in both CCS and SCS. It is interesting to observe that, $i^2 = \begin{bmatrix} 1 \\ \pi/2 \\ 0 \end{bmatrix} \begin{bmatrix} 1 \\ \pi/2 \\ 0 \end{bmatrix} = \begin{bmatrix} 1 \\ \pi \\ 0 \end{bmatrix} \Rightarrow i^2 = -1$. This behaves like a 2D negative unit multiplication

factor because for any $\begin{bmatrix} r \\ \phi \\ \theta \end{bmatrix}$, $\begin{bmatrix} r \\ \phi \\ \theta \end{bmatrix} \begin{bmatrix} 1 \\ \pi \\ 0 \end{bmatrix} = \begin{bmatrix} r \\ \phi + \pi \\ \theta \end{bmatrix} = -a - ib + jc$, i.e., it changes the

angle $\phi$ by $\pi$ rad and does not affect $\theta$. Moreover, $j^2 = \begin{bmatrix} 1 \\ \phi \\ \pi/2 \end{bmatrix} \begin{bmatrix} 1 \\ \phi \\ \pi/2 \end{bmatrix} = \begin{bmatrix} 1 \\ 2\phi \\ 0 \end{bmatrix} = -1$ if $\phi =$

$\pi/2$, and 1 if $\phi = 0$. Similarly, $ij = \begin{bmatrix} 1 \\ \pi/2 \\ 0 \end{bmatrix} \begin{bmatrix} 1 \\ \phi \\ \pi/2 \end{bmatrix} = \begin{bmatrix} 1 \\ \pi/2 + \phi \\ \pi/2 \end{bmatrix} \Rightarrow (ij)^2 = \begin{bmatrix} 1 \\ \pi + 2\phi \\ 0 \end{bmatrix} =$

$-1$ for $\phi = 0$, and $(ij)^2 = 1$ for $\phi = \pi/2$.

Further, if $g_1 = \begin{bmatrix} 1 \\ 0 \\ \theta_1 \end{bmatrix}$ and $g_2 = \begin{bmatrix} 1 \\ 0 \\ \theta_2 \end{bmatrix}$, then $g_1 g_2 = \begin{bmatrix} 1 \\ 0 \\ \theta_1 + \theta_2 \end{bmatrix}$ and in general,

$g_1^m g_2^n = \begin{bmatrix} 1 \\ 0 \\ m\theta_1 + n\theta_2 \end{bmatrix}$. In fact, the new imaginary number $j$ has one degree of freedom

because it can be written as $j = \begin{bmatrix} 1 \\ \phi \\ \pi/2 \end{bmatrix}$ for any $\phi \in [0, 2\pi)$ which implies $j^2 = \begin{bmatrix} 1 \\ 2\phi \\ 0 \end{bmatrix}$, and

thus

$$j^2 = \cos(2\phi) + i\sin(2\phi) \text{ for } \phi \in [0, 2\pi), \tag{30}$$

therefore, it can have infinite number of representations. For examples (i) $j^2 = 1$ when $\phi = 0$, (ii) $j^2 = -1$ when $\phi = \pi/2$, (iii) $j^2 = i$ when $\phi = \pi/4$, (iv) $j^2 = -i$ when $\phi = 3\pi/4$. Basically, the multiplication of a complex number, e.g., $\cos(\phi) + i\sin(\phi) = \begin{bmatrix} 1 \\ \phi \\ 0 \end{bmatrix}$ with specific $j = \begin{bmatrix} 1 \\ 0 \\ \pi/2 \end{bmatrix}$ is

another representation of $j = \begin{bmatrix} 1 \\ \phi \\ \pi/2 \end{bmatrix}$. We observe that these **infinite representations of** $j$

**arise solely from the nonunique representation of a point on the** $Z$**-axis in the SCS**, as elucidated in Remark 1. These nonuniqueness can be easily avoided by considering $\theta \in (-\pi/2, \pi/2)$ as explained in Remark 3.

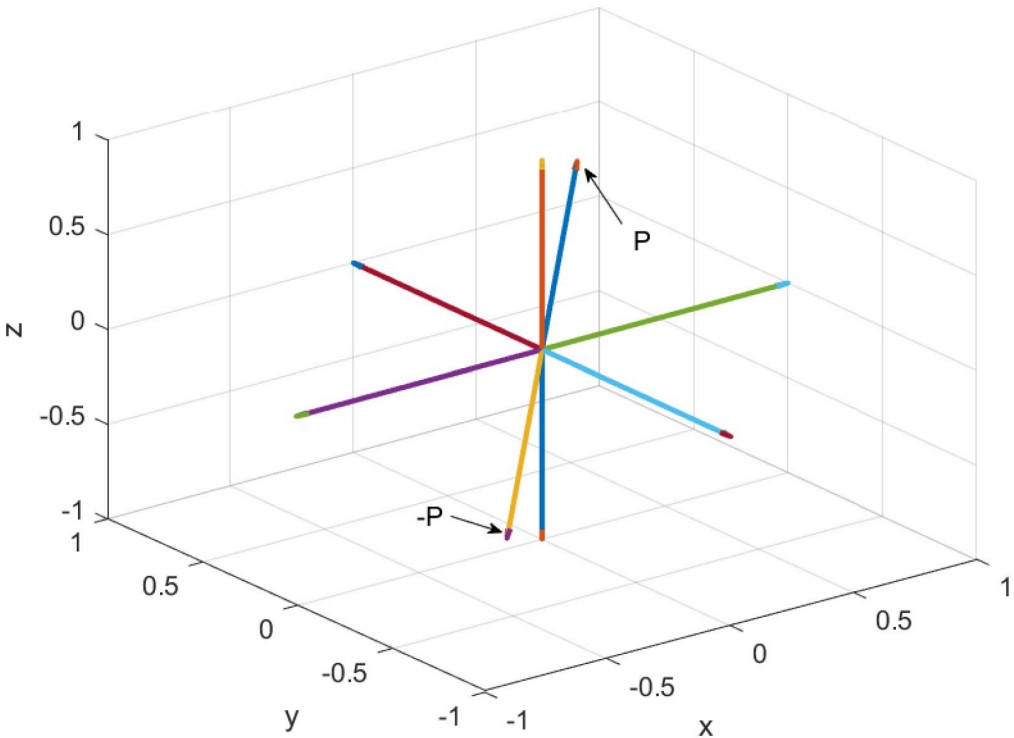

**Fig 6. A point** $P = \frac{1}{2\sqrt{2}}(\sqrt{3} + i\sqrt{3} + j\sqrt{2})$ **in the considered spherical co-ordinate system where radius $r = 1$, azimuth angle $\phi = \pi/4$ and elevation angle $\theta = \pi/6$ are shown, and point $-P$ with $r = 1$, $\phi = 5\pi/4$ and elevation angle $\theta = 7\pi/6 = -\pi/6$ are also shown.**

The additive inverse of an element $g = a + ib + jc$ is given by

$$g_{ai} = -a - ib - jc = \begin{bmatrix} -a \\ -b \\ -c \end{bmatrix} \simeq \begin{bmatrix} r \\ \phi + \pi \\ -\theta \end{bmatrix} = \begin{bmatrix} 1 \\ \pi \\ 0 \end{bmatrix} \begin{bmatrix} r \\ \phi \\ -\theta \end{bmatrix}, \tag{31}$$

and thus $g + g_{ai} = 0 \Rightarrow g_{ai} = -g$, e.g., a point $P$ and $-P$ is shown in Fig 6. It is important to note that $(-1)g \neq g_{ai}$ because multiplication by $-1$ changes the angle $\phi$ by $\pi$ rad but does not affect $\theta$, while we obtain $g_{ai}$ from $g$ by mapping $\phi \mapsto \phi + \pi$ and $\theta \mapsto -\theta$ as defined in (31), which is equivalent to multiplication by $-1$ and conjugating with respect to $j$ as defined in (19), i.e.,

$g_{ai} = (-1)\bar{g}_j$. Therefore $g\,g_{ai} = \begin{bmatrix} r \\ \phi \\ \theta \end{bmatrix} \begin{bmatrix} r \\ \phi + \pi \\ -\theta \end{bmatrix} = \begin{bmatrix} r^2 \\ 2\phi + \pi \\ 0 \end{bmatrix} \neq (-1)g^2;$

$(g_{ai})(g_{ai}) = \begin{bmatrix} r \\ \phi + \pi \\ -\theta \end{bmatrix} \begin{bmatrix} r \\ \phi + \pi \\ -\theta \end{bmatrix} = \begin{bmatrix} r^2 \\ 2\phi + 2\pi \\ -2\theta \end{bmatrix} = \begin{bmatrix} r^2 \\ 2\phi \\ -2\theta \end{bmatrix} = (g_{ai})^2 \neq g^2$, and

$(g_{ai})\bar{g} = \begin{bmatrix} r \\ \phi + \pi \\ -\theta \end{bmatrix} \begin{bmatrix} r \\ -\phi \\ -\theta \end{bmatrix} = \begin{bmatrix} r^2 \\ \pi \\ -2\theta \end{bmatrix}$. One can notice that the angle $\phi$ is wrapped in the multiple of $2\pi$ and $\theta$ in the multiple of $\pi$.

Because the above defined spherical multiplication is not derived from addition, it does not follow the distributive property, and thus, in general, $g_1(g_2 + g_3) \neq g_1 g_2 + g_1 g_3$, likewise addition

does not distribute over multiplication, i.e., $g_1 + (g_2 g_3) \neq (g_1 + g_2)(g_1 + g_3)$. This is a desired property of the defined SM because, geometrically, the operation on the left side is different from that on the right side.

**Remark 3**. As elucidated, when $\theta$ assumes values of $\pm \pi/2$, the azimuthal angle $\phi$ becomes indeterminate, leading to an infinite set of representations for the complex number $j$. This inherent indeterminacy underscores the existence of non-uniqueness in the representations. In order to establish complete uniqueness, we impose the constraint $\theta \in (-\pi/2, \pi/2)$, thereby excluding only two endpoints from the interval $\theta \in [-\pi/2, \pi/2]$. This restriction in (1) ensures that if $c \neq 0$, then either $a \neq 0$ or $b \neq 0$, but both $a$ and $b$ cannot be equal to zero simultaneously. Consequently, a practical assurance of uniqueness in the representation of numbers within the proposed system is attainable such that

$$\begin{bmatrix} a_1 \\ b_1 \\ c_1 \end{bmatrix} + \begin{bmatrix} a_2 \\ b_2 \\ c_2 \end{bmatrix} = \begin{bmatrix} a_1 + a_2 \\ b_1 + b_2 \\ c_1 + c_2 \end{bmatrix} = \begin{bmatrix} r_1 \cos\pi(\theta_1)\cos(\phi_1) + r_2 \cos\pi(\theta_2)\cos(\phi_2) \\ r_1 \cos\pi(\theta_1)\sin(\phi_1) + r_2 \cos\pi(\theta_2)\sin(\phi_2) \\ r_1 \sin\pi(\theta_1) + r_2 \sin\pi(\theta_2) \end{bmatrix}, \quad (32)$$

$$\text{and} \quad \begin{bmatrix} a_1 \\ b_1 \\ c_1 \end{bmatrix} \begin{bmatrix} a_2 \\ b_2 \\ c_2 \end{bmatrix} = \begin{bmatrix} r_1 r_2 \cos\pi(\theta_1 + \theta_2)\cos(\phi_1 + \phi_2) \\ r_1 r_2 \cos\pi(\theta_1 + \theta_2)\sin(\phi_1 + \phi_2) \\ r_1 r_2 \sin\pi(\theta_1 + \theta_2) \end{bmatrix} \simeq \begin{bmatrix} r_1 r_2 \\ \phi_1 + \phi_2 \\ \theta_1 + \theta_2 \end{bmatrix}, \quad (33)$$

where from (3) $r_1 = \sqrt{a_1^2 + b_1^2 + c_1^2}$, $r_2 = \sqrt{a_2^2 + b_2^2 + c_2^2}$, $\phi_1 = \arctan\left(\frac{b_1}{a_1}\right)$ $\phi_2 = \arctan\left(\frac{b_2}{a_2}\right)$, $\theta_1 = \arctan\left(\frac{c_1}{\sqrt{a_1^2 + b_1^2}}\right)$, $\theta_2 = \arctan\left(\frac{c_2}{\sqrt{a_2^2 + b_2^2}}\right)$, functions $\cos\pi(\cdot)$ and $\sin\pi(\cdot)$ are defined in (4) and (5), respectively.

Next, we present two important results as follows:

**Result 1**. Using the proposed theory with $j^2 = \pm 1 \Leftrightarrow j^3 = \pm j$ and $j^2 = \pm 1 \Rightarrow j^4 = 1$, for all $\theta \in \mathbb{R}$ with mapping $\theta \mapsto \theta \,\hat{\mathrm{m}}\mathrm{od}\, \pi$ as defined in (26), we obtain

$$e^{j\theta} = 1 + \frac{j\theta}{1!} + \frac{(j\theta)^2}{2!} + \frac{(j\theta)^3}{3!} + \frac{(j\theta)^4}{4!} + \frac{(j\theta)^5}{5!} + \cdots + \frac{(j\theta)^n}{n!} + \cdots \quad (34)$$

On using $j^2 = -1$, $j^3 = -j$ and $j^4 = 1$, we can easily obtain Euler identity as

$$\begin{aligned} e^{j\theta} &= \left[ 1 - \frac{\theta^2}{2!} + \frac{\theta^4}{4!} - \frac{\theta^6}{6!} + \cdots \right] + j \left[ \frac{\theta}{1!} - \frac{\theta^3}{3!} + \frac{\theta^5}{5!} - \frac{\theta^7}{7!} + \cdots \right], \\ &= \cos(\theta) + j \sin(\theta), \\ &= \cos\pi(\theta) + j \sin\pi(\theta), \quad \text{as } \theta \mapsto \theta \,\hat{\mathrm{m}}\mathrm{od}\, \pi \text{ from } (26). \end{aligned} \quad (35)$$

Interestingly, on using $j^2 = 1$, $j^3 = j$ and $j^4 = 1$, we obtain hyperbolic Euler type identity as

$$\begin{aligned} e_h^{j\theta} &= \left[ 1 + \frac{\theta^2}{2!} + \frac{\theta^4}{4!} + \frac{\theta^6}{6!} + \cdots \right] + j \left[ \frac{\theta}{1!} + \frac{\theta^3}{3!} + \frac{\theta^5}{5!} + \frac{\theta^7}{7!} + \cdots \right], \\ &= \cosh(\theta) + j \sinh(\theta). \end{aligned} \quad (36)$$

We observe that $e^{j\theta} = e^{j(\theta + n\pi)} = e^{j(\theta \,\hat{\mathrm{m}}\mathrm{od}\, \pi)}$ for all $n \in \mathbb{Z}$, indicating that $e^{j\theta}$ is $\pi$-periodic. This behavior is analogous to $e^{i\phi} = e^{i(\phi + 2n\pi)} = e^{i(\phi \,\mathrm{mod}\, 2\pi)}$, which exhibits $2\pi$-periodicity. In both

cases, the periodic nature of the exponential function arises from the fundamental periodicity of the complex exponential, with the periodicity determined by the coefficient of the argument $\pi$ or $2\pi$, respectively. Moreover, from (4) and (5)

$$\cos\pi(\theta) = \begin{cases} \mathrm{Re}(e^{j\theta}) = \sum_{n=0}^{\infty}(-1)^n \dfrac{(\theta)^{2n}}{(2n)!}, & \text{if } \theta \in \bigcup_{n=-\infty}^{\infty}\left[-\dfrac{\pi}{2}+2\pi n, \dfrac{\pi}{2}+2\pi n\right], \\[3mm] \mathrm{Re}(-e^{j\theta}) = -\sum_{n=0}^{\infty}(-1)^n \dfrac{(\theta)^{2n}}{(2n)!}, & \text{if } \theta \in \bigcup_{n=-\infty}^{\infty}\left(\dfrac{\pi}{2}+2\pi n, \dfrac{3\pi}{2}+2\pi n\right), \end{cases}$$

and $\quad\sin\pi(\theta) = \begin{cases} \mathrm{Im}(e^{j\theta}) = \sum_{n=0}^{\infty}(-1)^n \dfrac{(\theta)^{2n+1}}{(2n+1)!}, & \text{if } \theta \in \bigcup_{n=-\infty}^{\infty}\left[-\dfrac{\pi}{2}+2\pi n, \dfrac{\pi}{2}+2\pi n\right], \\[3mm] \mathrm{Im}(-e^{j\theta}) = -\sum_{n=0}^{\infty}(-1)^n \dfrac{(\theta)^{2n+1}}{(2n+1)!}, & \text{if } \theta \in \bigcup_{n=-\infty}^{\infty}\left(\dfrac{\pi}{2}+2\pi n, \dfrac{3\pi}{2}+2\pi n\right). \end{cases}$

**Example 1**. Now, we consider an interesting example that will be useful in obtaining next result as follows: Let $g_1 = \begin{bmatrix} r_1 \\ \phi \\ 0 \end{bmatrix}$ and $g_2 = \begin{bmatrix} r_2 \\ 0 \\ \theta \end{bmatrix}$, where $r_1, r_2 > 0$. Using new multiplication defined in (16) we obtain $g_1 g_2 = \begin{bmatrix} r_1 \\ \phi \\ 0 \end{bmatrix}\begin{bmatrix} r_1 \\ 0 \\ \theta \end{bmatrix} = \begin{bmatrix} r_1 r_2 \\ \phi \\ \theta \end{bmatrix}$. Using (1) to (14), this can be written as $g_1 g_2 = r_1 r_2[\cos\pi(\theta)\cos(\phi) + i\cos\pi(\theta)\sin(\phi) + j\sin\pi(\theta)]$, and thus,

$$re^{i\phi}e^{j\theta} = r(\cos\pi(\theta)\cos(\phi) + i\cos\pi(\theta)\sin(\phi) + j\sin\pi(\theta)). \tag{37}$$

Moreover, $g_1^m = r_1^m e^{im\phi}$ and $g_2^m = r_2^m e^{jm\theta}$, thus the behaviour of these two complex numbers is very similar.

**Example 2**. Consider two complex numbers $g_1 = r_1 e^{i\phi_1}e^{j\theta_1}$ and $g_2 = r_2 e^{i\phi_2}e^{j\theta_2}$. The sum of these two complex numbers is expressed as $g_1 + g_2 = r_1 e^{i\phi_1}e^{j\theta_1} + r_2 e^{i\phi_2}e^{j\theta_2}$. Now, let us explore two distinct scenarios:

(i) When $r_1 = r_2 = r$ and $\phi_1 = \phi_2 = \phi$ implying $g_1 + g_2 = re^{i\phi}e^{j\theta_1} + re^{i\phi}e^{j\theta_2} = r(\cos\pi(\theta_1)\cos(\phi) + i\cos\pi(\theta_1)\sin(\phi) + j\sin\pi(\theta_1)) + r(\cos\pi(\theta_2)\cos(\phi) + i\cos\pi(\theta_2)\sin(\phi) + j\sin\pi(\theta_2))$. By simplifying this expression, we obtain: $g_1 + g_2 = r[\cos(\phi)(\cos\pi(\theta_1) + \cos\pi(\theta_2)) + i\sin(\phi)(\cos\pi(\theta_1) + \cos\pi(\theta_2)) + j(\sin\pi(\theta_1) + \sin\pi(\theta_2))] = r[(\cos(\phi) + i\sin(\phi))(\cos\pi(\theta_1) + \cos\pi(\theta_2)) + j(\sin\pi(\theta_1) + \sin\pi(\theta_2))] = r[e^{i\phi}(\cos\pi(\theta_1) + \cos\pi(\theta_2)) + j(\sin\pi(\theta_1) + \sin\pi(\theta_2))] = re^{i\phi}[(\cos\pi(\theta_1) + \cos\pi(\theta_2)) + j(\sin\pi(\theta_1) + \sin\pi(\theta_2))] = re^{i\phi}[e^{j\theta_1} + e^{j\theta_2}]$. Here, we have utilized the fact that $j(\sin\pi(\theta_1) + \sin\pi(\theta_2)) \times e^{i\phi} = j(\sin\pi(\theta_1) + \sin\pi(\theta_2))$.

(ii) In the case where $r_1 = r_2 = r$ and $\theta_1 = \theta_2 = \theta$, we find that $g_1 + g_2 = re^{i\phi_1}e^{j\theta} + re^{i\phi_2}e^{j\theta} = r(\cos\pi(\theta)\cos(\phi_1) + i\cos\pi(\theta)\sin(\phi_1) + j\sin\pi(\theta)) + r(\cos\pi(\theta)\cos(\phi_2) + i\cos\pi(\theta)\sin(\phi_2) + j\sin\pi(\theta))$. Simplifying further, we obtain: $g_1 + g_2 = r[\cos\pi(\theta)(\cos(\phi_1) + \cos(\phi_2)) + i\cos\pi(\theta)(\sin(\phi_1) + \sin(\phi_2)) + j2\sin\pi(\theta)] \neq re^{j\theta}(e^{i\phi_1} + e^{i\phi_2}) = re^{j\theta}[(\cos(\phi_1) + \cos(\phi_2)) + i(\sin(\phi_1) + \sin(\phi_2))]$.

From these two cases, it is evident that:

$$re^{i\phi}e^{j\theta_1} + re^{i\phi}e^{j\theta_2} = re^{i\phi}(e^{j\theta_1} + e^{j\theta_2}),\tag{38}$$

$$\text{and} \quad re^{i\phi_1}e^{j\theta} + re^{i\phi_2}e^{j\theta} \neq re^{j\theta}(e^{i\phi_1} + e^{i\phi_2}).\tag{39}$$

Eqs (38) and (39) illustrate the distinction between the sum of complex numbers in the given cases, where $e^{i\phi}$ distributes while $e^{j\theta}$ does not distribute.

**Example 3**. We consider 3D hypercomplex numbers $g_1 = \begin{bmatrix} r_1 \\ \phi_1 \\ \theta_1 \end{bmatrix}$ and $g_2 = \begin{bmatrix} r_2 \\ \phi_2 \\ \theta_2 \end{bmatrix}$, where

$r_1, r_2 > 0$. Their respective additive inverses are given by $-g_1 = \begin{bmatrix} r_1 \\ \phi_1 + \pi \\ -\theta_1 \end{bmatrix}$ and

$-g_2 = \begin{bmatrix} r_2 \\ \phi_2 + \pi \\ -\theta_2 \end{bmatrix}$. Under the multiplication operation defined in (16), we obtain the following

results: $g_1 g_2 = \begin{bmatrix} r_1 \\ \phi_1 \\ \theta_1 \end{bmatrix}\begin{bmatrix} r_2 \\ \phi_2 \\ \theta_2 \end{bmatrix} = \begin{bmatrix} r_1 r_2 \\ \phi_1 + \phi_2 \\ \theta_1 + \theta_2 \end{bmatrix}$, $(-g_1)g_2 = \begin{bmatrix} r_1 \\ \phi_1 + \pi \\ -\theta_1 \end{bmatrix}\begin{bmatrix} r_2 \\ \phi_2 \\ \theta_2 \end{bmatrix} = \begin{bmatrix} r_1 r_2 \\ \phi_1 + \phi_2 + \pi \\ -\theta_1 + \theta_2 \end{bmatrix}$,

$g_1(-g_2) = \begin{bmatrix} r_1 \\ \phi_1 \\ \theta_1 \end{bmatrix}\begin{bmatrix} r_2 \\ \phi_2 + \pi \\ -\theta_2 \end{bmatrix} = \begin{bmatrix} r_1 r_2 \\ \phi_1 + \phi_2 + \pi \\ \theta_1 - \theta_2 \end{bmatrix}$, and $(-g_1)(-g_2) = \begin{bmatrix} r_1 \\ \phi_1 + \pi \\ -\theta_1 \end{bmatrix}\begin{bmatrix} r_2 \\ \phi_2 + \pi \\ -\theta_2 \end{bmatrix} =$

$\begin{bmatrix} r_1 r_2 \\ \phi_1 + \phi_2 + 2\pi \\ -\theta_1 - \theta_2 \end{bmatrix} = \begin{bmatrix} r_1 r_2 \\ \phi_1 + \phi_2 \\ -\theta_1 - \theta_2 \end{bmatrix}$. These expressions demonstrate that the multiplication

$(-g_1)g_2 \neq g_1(-g_2)$, and $g_1 g_2 \neq (-g_1)(-g_2)$. Further observations reveal that the multiplication

exhibits standard properties: $(-1)g_1 g_2 = [(-1)g_1]g_2 = g_1[(-1)g_2] = \begin{bmatrix} r_1 r_2 \\ \phi_1 + \phi_2 + \pi \\ \theta_1 + \theta_2 \end{bmatrix}$, and

$[(-1)g_1][(-1)g_2] = \begin{bmatrix} r_1 r_2 \\ \phi_1 + \phi_2 + 2\pi \\ \theta_1 + \theta_2 \end{bmatrix} = \begin{bmatrix} r_1 r_2 \\ \phi_1 + \phi_2 \\ \theta_1 + \theta_2 \end{bmatrix} = g_1 g_2$. These findings highlight the

non-trivial nature of multiplication within this algebraic structure.

Using these observations we present the following result.

**Result 2**. The distributive property of the defined spherical multiplication over addition (i.e., $g_1(g_2 + g_3) = g_1 g_2 + g_1 g_3$) holds if $g_1 \in \mathbb{S}^2$, and $g_2, g_3 \in \mathbb{S}^3$. Using (38) we easily obtain

$$\begin{bmatrix} r_1 \\ \phi_1 \\ 0 \end{bmatrix} \cdot \left( \begin{bmatrix} r_2 \\ \phi_2 \\ \theta_2 \end{bmatrix} + \begin{bmatrix} r_3 \\ \phi_3 \\ \theta_3 \end{bmatrix} \right) = \begin{bmatrix} r_1 r_2 \\ \phi_1 + \phi_2 \\ \theta_2 \end{bmatrix} + \begin{bmatrix} r_1 r_3 \\ \phi_1 + \phi_3 \\ \theta_3 \end{bmatrix}.$$

**Result 3**. Consider a 3D hypercomplex number $g = a + ib + jc$, where $a, b, c$ are real num-

bers. It follows that the exponential of $g$ is given by: $e^g = e^a e^{ib} e^{jc} = \begin{bmatrix} e^a \\ 0 \\ 0 \end{bmatrix}\begin{bmatrix} 1 \\ b \\ 0 \end{bmatrix}\begin{bmatrix} 1 \\ 0 \\ c \end{bmatrix} =$

$$\begin{bmatrix} e^a \\ b \\ c \end{bmatrix} = e^a[\cos(b)\cos\pi(c) + i\sin(b)\cos\pi(c) + j\sin\pi(c)].$$ The natural logarithm of $e^g$ is given

by: $\ln(e^g) = g \Rightarrow \ln\left(\begin{bmatrix} e^a \\ b \\ c \end{bmatrix}\right) = a + ib + jc$. Therefore, for any 3D hypercomplex number

$g = a + ib + jc \Rightarrow g = re^{i\phi}e^{j\theta} \simeq \begin{bmatrix} r \\ \phi \\ \theta \end{bmatrix}$, we obtain

$$\begin{aligned} \ln(g) &= \ln(r) + i\phi + j\theta, \quad \text{and} \\ e^g &= e^a[\cos(b)\cos\pi(c) + i\sin(b)\cos\pi(c) + j\sin\pi(c)]. \end{aligned} \tag{40}$$

The expression (40) reduces to the conventional 2D complex number system if $c = 0$, leading to $\theta = 0$. This result establishes a fundamental relationship between the exponential, logarithmic, and spherical polar forms of 3D hypercomplex numbers, extending the concepts from complex analysis to a higher-dimensional setting. The spherical polar representation offers a compact and geometric way to visualize and manipulate 3D hypercomplex numbers, analogous to the use of polar coordinates for complex numbers.

Thus, the proposed 3D hypercomplex number system is a true generalization of the existing 2D complex number system. To obtain the multiplication of two numbers, we can use *Result 3* as follows: Let $\ln(g_1) = \ln(r_1) + i\phi_1 + j\theta_1$ and $\ln(g_2) = \ln(r_2) + i\phi_2 + j\theta_2$, and thus

$\ln(g_1) + \ln(g_2) = \ln(g_1 g_2) = \ln(r_1 r_2) + i(\phi_1 + \phi_2) + j(\theta_1 + \theta_2) \Rightarrow g_1 g_2 = \begin{bmatrix} r_1 r_2 \\ \phi_1 + \phi_2 \\ \theta_1 + \theta_2 \end{bmatrix}$. There-

fore, we conclude that the addition of hypercomplex numbers is naturally defined in the Cartesian coordinates and *multiplication is naturally defined in the spherical coordinates through the natural logarithmic addition.*

Building upon the constrained periodicity of the hypercomplex number in (7) and the unifying multiplication framework established in (16), we present a novel non-distributive normed division algebra that extends its reach to dimensions previously considered inaccessible. This advancement marks a significant milestone in the field, effectively addressing a longstanding challenge and providing a comprehensive framework for encompassing non-distributive normed division algebras across all dimensionalities. Crucially, the defined spherical multiplication (16) demonstrates seamless backward compatibility with the established complex number multiplication, thereby serving as a generalization of this fundamental operation to the realm of higher-dimensional hypercomplex number systems. To elucidate this compatibility and its broader implications, using the above representations, we present the subsequent results.

**Theorem 1**. *A non-distributive normed division algebra (NDF) is a number system equipped with the fundamental arithmetic operations of addition, subtraction, multiplication, and division. It notably satisfies the norm condition $\|g_1 g_2\| = \|g_1\|\|g_2\|$ and has a dimension of $M = 3$. Furthermore, its algebraic structure exhibits distributivity for $M = 1$ and $M = 2$.*

*Proof.* To establish the theorem, we proceed to demonstrate that the 3D numbers, defined in (1) and (7), constitute a non-distributive field under the operations of addition (15) and multiplication (16). We meticulously verify the following axioms for all elements $g_1, g_2, g_3 \in \mathbb{S}^3$, as specified in (14):

1. Closure of addition and multiplication: For all $g_1, g_2 \in \mathbb{S}^3$, it holds that $g_1 + g_2 =$

$(a_1 + a_2) + i(b_1 + b_2) + j(c_1 + c_2) \in \mathbb{S}^3$, and $g_1 \cdot g_2 = \begin{bmatrix} r_1 r_2 \\ \phi_1 + \phi_2 \\ \theta_1 + \theta_2 \end{bmatrix} \in \mathbb{S}^3$. This is evident

from the definitions of addition (15) and multiplication (16).

2. Associativity of addition and multiplication: Both $g_1 + (g_2 + g_3) = (g_1 + g_2) + g_3$ and $g_1 \cdot (g_2 \cdot$

$g_3) = (g_1 \cdot g_2) \cdot g_3 = \begin{bmatrix} r_1 r_2 r_3 \\ \phi_1 + \phi_2 + \phi_3 \\ \theta_1 + \theta_2 + \theta_3 \end{bmatrix}$ yield expressions in $\mathbb{S}^3$ and are evident from (15)

and (16).

3. Commutativity of addition and multiplication: The commutative natures of $g_1 + g_2 = g_2 +$

$g_1$ and $g_1 \cdot g_2 = g_2 \cdot g_1 = \begin{bmatrix} r_1 r_2 \\ \phi_1 + \phi_2 \\ \theta_1 + \theta_2 \end{bmatrix}$ are apparent from (15) and (16).

4. Additive and multiplicative identities: For every $g \in \mathbb{S}^3$, there exist distinct elements 0 and
   1 in $\mathbb{S}^3$, as defined in (28) and (29), such that $g + 0 = g$ and $g \cdot 1 = g$.

5. Additive and multiplicative inverses: For every $g \in \mathbb{S}^3$, there exists an element $g_{ai} \in \mathbb{S}^3$
   called the additive inverse of $g$, such that $g + g_{ai} = 0$ where $g = a + ib + jc$ and $g_{ai} = -a - ib - jc$. For every nonzero element $g \in \mathbb{S}^3$, there exists an element $g^{-1}$ or $1/g$ in $\mathbb{S}^3$ called the mul-

   tiplicative inverse of $g$, such that $g \cdot g^{-1} = 1$. Here, $g^{-1} = \frac{\bar{g}}{g\bar{g}} = \frac{a-ib-jc}{a^2+b^2+c^2} = \begin{bmatrix} 1/r \\ -\phi \\ -\theta \end{bmatrix}$ as defined

   in (20), and is unique.

6. Distributivity of multiplication over addition: Generally, this property does not hold uni-
   versally in $\mathbb{S}^3$ as $g_1 \cdot (g_2 + g_3) \neq (g_1 \cdot g_2) + (g_1 \cdot g_3)$ or

   $\begin{bmatrix} r_1 \\ \phi_1 \\ \theta_1 \end{bmatrix} \cdot \left( \begin{bmatrix} r_2 \\ \phi_2 \\ \theta_2 \end{bmatrix} + \begin{bmatrix} r_3 \\ \phi_3 \\ \theta_3 \end{bmatrix} \right) \neq \begin{bmatrix} r_1 r_2 \\ \phi_1 + \phi_2 \\ \theta_1 + \theta_2 \end{bmatrix} + \begin{bmatrix} r_1 r_3 \\ \phi_1 + \phi_3 \\ \theta_1 + \theta_3 \end{bmatrix}$. Specifically, if $g_1 \in \mathbb{S}^2$ (imply-

   ing $\theta_1 = 0$), distributivity is observed in the representation as

   $\begin{bmatrix} r_1 \\ \phi_1 \\ 0 \end{bmatrix} \cdot \left( \begin{bmatrix} r_2 \\ \phi_2 \\ \theta_2 \end{bmatrix} + \begin{bmatrix} r_3 \\ \phi_3 \\ \theta_3 \end{bmatrix} \right) = \begin{bmatrix} r_1 r_2 \\ \phi_1 + \phi_2 \\ \theta_2 \end{bmatrix} + \begin{bmatrix} r_1 r_3 \\ \phi_1 + \phi_3 \\ \theta_3 \end{bmatrix}$.

   Having rigorously verified all the aforementioned axioms, except for general distributivity,
   we conclusively establish that the proposed 3D numbers indeed constitute a non-distributive
   field.

   **Remark 4**. We can observe that the proposed set $(\mathbb{S}^3, +)$ is an additive Abelian group, and
   $(\mathbb{S}^3 \setminus \{0\}, \cdot)$ is a multiplicative normed Abelian group (NAG). The set $(\mathbb{S}^3 \setminus \{0\}, \cdot)$ satisfies
   the fundamental group axioms along with the preservation of the norm, $\forall g, g_1, g_2, g_3 \in \mathbb{S}^3$, as

follows: (i) closure: $g_1 \cdot g_2 \in \mathbb{S}^3$, (ii) associativity: $(g_1 \cdot g_2) \cdot g_3 = g_1 \cdot (g_2 \cdot g_3)$, (iii) identity: $1 \cdot g = g$, (iv) inverses: $g \cdot g^{-1} = 1$, (v) commutativity: $g_1 \cdot g_2 = g_2 \cdot g_1$, and (vi) preservation of the norm: $\|g_1 \cdot g_2\| = \|g_1\|\|g_2\|$.

**Proposition 1**. *The following elementary consequences of the field axioms are also being satisfied by the proposed number systems*, $\forall g, g_1, g_2, g_3 \in \mathbb{S}^3$

1. $(-1)[(-1)g] = g$

2. $(g^{-1})^{-1} = g$

3. $g_1 + g_2 = g_1 + g_3 \Rightarrow g_2 = g_3$

4. $g0 = 0$

5. $[(-1)g_1]g_2 = (-1)(g_1 g_2)$

6. $[(-1)g_1][(-1)g_2] = g_1 g_2$

7. $g_1 g_2 = g_1 g_3$ *and* $g_1 \neq 0$ *implies* $g_2 = g_3$

8. $g_1 g_2 = 0 \Rightarrow g_1 = 0$ or $g_2 = 0$.

## 3.2 Examples of the proposed 3D hypercomplex numbers: Roots computation

To initiate the analysis, we consider the computation of $n$-th roots for a given polynomial,

$g^n = g_0 = \begin{bmatrix} r_0 \\ \phi_0 \\ \theta_0 \end{bmatrix}$, using (21). The resulting $n$-th root is expressed as $g = \begin{bmatrix} r_0^{1/n} \\ (\phi_0 + 2\pi k)/n \\ (\theta_0 + \pi \ell)/n \end{bmatrix}$,

where $k, \ell = 0, 1, \cdots, n-1$. This computational approach yields $n^2$ roots in $\mathbb{S}^3$, accounting for multiplicity. The systematic enumeration of these roots is accomplished by varying the parameters $k$ and $\ell$ within the specified ranges $k = 0, 1, \cdots, n-1$ and $\ell = 0, 1, \cdots, n-1$. Notably, the azimuth angle $\phi$ undergoes a modulo operation with respect to $2\pi$, while the elevation angle $\theta$ is subject to a modulo operation of $\pi$. This mathematical treatment is essential for a comprehensive examination of the solution space, ensuring a thorough coverage of the entire domain within the prescribed intervals. The resulting set of roots manifests diverse configurations within the confines of the 3D hypercomplex numbers in $\mathbb{S}^3$, yielding a total of $n^2$ solutions when $n$ is odd and, a total of $n^2$ or $n^2 + n$ solutions when $n$ is even. A presentation of illustrative examples is subsequently provided to elucidate the implications of this computational procedure.

**Example 4**. For example, let us consider the quadratic equation (QE) $x^2 + 1 = 0$. If $x \in \mathbb{R}$, then there are no real roots. If $x \in \mathbb{S}^2$ (traditionally, $x \in \mathbb{C}$), then there are two roots $x = \pm i$ where $i^2 = -1$. If $x \in \mathbb{S}^3$, then there are four roots as $x^2 = e^{i(\pi + 2\pi k)} e^{j(\pi l)} \Rightarrow x = e^{i(\pi + 2\pi k)/2} e^{j(\pi l)/2}$,

for $k, l = 0, 1$. Therefore, four roots are $\begin{bmatrix} 1 \\ \pi/2 \\ 0 \end{bmatrix}, \begin{bmatrix} 1 \\ 3\pi/2 \\ 0 \end{bmatrix}; \begin{bmatrix} 1 \\ \pi/2 \\ \pi/2 \end{bmatrix}, \begin{bmatrix} 1 \\ 3\pi/2 \\ \pi/2 \end{bmatrix}$ where last two

roots represent the same point in $Z$-axis. The other two roots are in the same $Z$-axis as

$\begin{bmatrix} 1 \\ \pi/2 \\ -\pi/2 \end{bmatrix}$ and $\begin{bmatrix} 1 \\ 3\pi/2 \\ -\pi/2 \end{bmatrix}$ which represent the same point in $Z$-axis. So there are total 6 roots of

unity in 3D complex number system. However, it is worth to note that the polynomial, $x^2 + 1 = 0$, has infinite number of quaternion roots.

**Example 5**. Here, we consider $x^3 - 1 = 0$, where $x \in \mathbb{S}^3$ and compute its roots as $x^3 = e^{i2\pi k}e^{j\pi l}$

$\Rightarrow x = e^{i2\pi k/3}e^{j\pi l/3}$, where $k, l = 0, 1, 2$. Therefore, roots are $\begin{bmatrix} 1 \\ 0 \\ 0 \end{bmatrix}, \begin{bmatrix} 1 \\ 2\pi/3 \\ 0 \end{bmatrix}, \begin{bmatrix} 1 \\ 4\pi/3 \\ 0 \end{bmatrix}$;

$\begin{bmatrix} 1 \\ 0 \\ \pi/3 \end{bmatrix}, \begin{bmatrix} 1 \\ 2\pi/3 \\ \pi/3 \end{bmatrix}, \begin{bmatrix} 1 \\ 4\pi/3 \\ \pi/3 \end{bmatrix}; \begin{bmatrix} 1 \\ 0 \\ 2\pi/3 \end{bmatrix}, \begin{bmatrix} 1 \\ 2\pi/3 \\ 2\pi/3 \end{bmatrix}, \begin{bmatrix} 1 \\ 4\pi/3 \\ 2\pi/3 \end{bmatrix}$ where $2\pi/3$ is same as $2\pi/3 - \pi = -\pi/3$ for angle $\theta$. So there are 9 distinct roots of unity in 3D complex number system.

**Example 6**. Here, we consider $x^4 - 1 = 0$, where $x \in \mathbb{S}^3$ and compute its roots as $x^4 = e^{i2\pi k}e^{j\pi l}$

$\Rightarrow x = e^{i\pi k/2}e^{j\pi l/4}$, where $k, l = 0, 1, 2, 3$. Therefore, roots are $\begin{bmatrix} 1 \\ 0 \\ 0 \end{bmatrix}, \begin{bmatrix} 1 \\ \pi/2 \\ 0 \end{bmatrix}, \begin{bmatrix} 1 \\ \pi \\ 0 \end{bmatrix}, \begin{bmatrix} 1 \\ 3\pi/2 \\ 0 \end{bmatrix}$;

$\begin{bmatrix} 1 \\ 0 \\ \pi/4 \end{bmatrix}, \begin{bmatrix} 1 \\ \pi/2 \\ \pi/4 \end{bmatrix}, \begin{bmatrix} 1 \\ \pi \\ \pi/4 \end{bmatrix}, \begin{bmatrix} 1 \\ 3\pi/2 \\ \pi/4 \end{bmatrix}$;

$\begin{bmatrix} 1 \\ 0 \\ \pi/2 \end{bmatrix}, \begin{bmatrix} 1 \\ \pi/2 \\ \pi/2 \end{bmatrix}, \begin{bmatrix} 1 \\ \pi \\ \pi/2 \end{bmatrix}, \begin{bmatrix} 1 \\ 3\pi/2 \\ \pi/2 \end{bmatrix}; \begin{bmatrix} 1 \\ 0 \\ 3\pi/4 \end{bmatrix}, \begin{bmatrix} 1 \\ \pi/2 \\ 3\pi/4 \end{bmatrix}, \begin{bmatrix} 1 \\ \pi \\ 3\pi/4 \end{bmatrix}, \begin{bmatrix} 1 \\ 3\pi/2 \\ 3\pi/4 \end{bmatrix}$ where $3\pi/4$

is same as $3\pi/4 - \pi = -\pi/4$ for angle $\theta$. Other four roots are $\begin{bmatrix} 1 \\ 0 \\ -\pi/2 \end{bmatrix}, \begin{bmatrix} 1 \\ \pi/2 \\ -\pi/2 \end{bmatrix}, \begin{bmatrix} 1 \\ \pi \\ -\pi/2 \end{bmatrix}$,

$\begin{bmatrix} 1 \\ 3\pi/2 \\ -\pi/2 \end{bmatrix}$. So there are total 20 roots in 3D complex number system.

**Example 7**. Here, we consider $x^4 + 1 = 0$, where $x \in \mathbb{S}^3$ and compute its roots as $x^4 = e^{i(\pi + 2\pi k)}e^{j(\pi l)} \Rightarrow x = e^{i(\pi + 2\pi k)/4}e^{j(\pi l)/4}$, where $k, l = 0, 1, 2, 3$. Therefore, roots are

$\begin{bmatrix} 1 \\ \pi/4 \\ 0 \end{bmatrix}, \begin{bmatrix} 1 \\ 3\pi/4 \\ 0 \end{bmatrix}, \begin{bmatrix} 1 \\ 5\pi/4 \\ 0 \end{bmatrix}, \begin{bmatrix} 1 \\ 7\pi/4 \\ 0 \end{bmatrix}; \begin{bmatrix} 1 \\ \pi/4 \\ \pi/4 \end{bmatrix}, \begin{bmatrix} 1 \\ 3\pi/4 \\ \pi/4 \end{bmatrix}, \begin{bmatrix} 1 \\ 5\pi/4 \\ \pi/4 \end{bmatrix}, \begin{bmatrix} 1 \\ 7\pi/4 \\ \pi/4 \end{bmatrix}$;

$\begin{bmatrix} 1 \\ \pi/4 \\ \pi/2 \end{bmatrix}, \begin{bmatrix} 1 \\ 3\pi/4 \\ \pi/2 \end{bmatrix}, \begin{bmatrix} 1 \\ 5\pi/4 \\ \pi/2 \end{bmatrix}, \begin{bmatrix} 1 \\ 7\pi/4 \\ \pi/2 \end{bmatrix}; \begin{bmatrix} 1 \\ \pi/4 \\ 3\pi/4 \end{bmatrix}, \begin{bmatrix} 1 \\ 3\pi/4 \\ 3\pi/4 \end{bmatrix}, \begin{bmatrix} 1 \\ 5\pi/4 \\ 3\pi/4 \end{bmatrix}, \begin{bmatrix} 1 \\ 7\pi/4 \\ 3\pi/4 \end{bmatrix}$. Other four

roots are $\begin{bmatrix} 1 \\ \pi/4 \\ -\pi/2 \end{bmatrix}, \begin{bmatrix} 1 \\ 3\pi/4 \\ -\pi/2 \end{bmatrix}, \begin{bmatrix} 1 \\ 5\pi/4 \\ -\pi/2 \end{bmatrix}, \begin{bmatrix} 1 \\ 7\pi/4 \\ -\pi/2 \end{bmatrix}$, so there are total 20 roots.

We can conclude the above observation as follows:

**Result 4**. The number of $n$-th roots of unity in $x \in \mathbb{S}^3$ are (i) $n^2$ if $n$ is an odd, and (ii) $n^2 + n$ if $n$ is an even number.

### 3.3 Geometrical insights into the generalized hypercomplex number system

We note that algebraically, the additional imaginary axis $j$ considered in $\mathbb{S}^{\mathbb{M}}$ behaves similar to $i$. For example, $i^2 = -1$ and $j^2 = -1$. Similarly, one can also show that $(1 + j)^2 = 2j$ and $(1 - j)^2 = -2j$. Similar identities are satisfied by $i$. Moreover, this $j$ axis geometrically plays interesteingly on the hypercomplex numbers. If there is a point $P = a + ib = \begin{bmatrix} r \\ \phi \\ 0 \end{bmatrix}$ in the complex $XY$ plane and if it is multiplied by a unit norm complex number $\begin{bmatrix} 1 \\ \varphi \\ 0 \end{bmatrix}$, then that point will rotate coun-terclockwise by $\varphi$, i.e., new point $Q = re^{i(\phi+\varphi)} = \begin{bmatrix} r \\ \phi + \varphi \\ 0 \end{bmatrix}$. Similarly, if a 3D point $P = a + ib + jc \Rightarrow P = \begin{bmatrix} r \\ \phi \\ \theta \end{bmatrix}$ is multiplied by a unit norm point $\begin{bmatrix} 1 \\ \varphi \\ \theta \end{bmatrix}$, then it will rotate to new point $Q = \begin{bmatrix} r \\ \phi + \varphi \\ \theta + \theta \end{bmatrix}$. Thus in the proposed 3D hypercomplex number system, one can rotate a point in both $\phi$ and $\theta$ directions with desired azimuth and elevation angles.

## 4 Proposed generalized (*M*D) hypercomplex number system

The 3D hypercomplex number system can be easily generalized to the $M$D hypercomplex number $\mathbb{S}^M$ system by using the generalized $M$D spherical coordinate system. For example, 4D hypercomplex number system can be written, for all $\phi_1 \in [0, 2\pi)$ and $\phi_2, \phi_3 \in [-\pi/2, \pi/2]$, as

$$
\begin{aligned}
d_0 &= r \cos\pi(\phi_3) \, \cos\pi(\phi_2)\cos(\phi_1), \\
d_1 &= r \cos\pi(\phi_3) \, \cos\pi(\phi_2)\sin(\phi_1), \\
d_2 &= r \cos\pi(\phi_3) \, \sin\pi(\phi_2), \\
d_3 &= r \sin\pi(\phi_3), \\
\phi_1 &= \arctan\left(\frac{d_1}{d_0}\right), \quad \phi_2 = \arctan\left(\frac{d_2}{\sqrt{d_0^2 + d_1^2}}\right), \quad \phi_3 = \arctan\left(\frac{d_3}{\sqrt{d_0^2 + d_1^2 + d_2^2}}\right), \\
r &= \sqrt{d_0^2 + d_1^2 + d_2^2 + d_3^2}, \quad g = d_0 + j_1 d_1 + j_2 d_2 + j_3 d_3,
\end{aligned}
\tag{41}
$$

where $j_2$ has one degree of freedom ($\phi_1$), and $j_3$ has two degree of freedom ($\phi_1$ and $\phi_2$). In

general, with $\phi_1 \in [0, 2\pi)$ and $\phi_2, \phi_3, \cdots, \phi_{M-1} \in [-\pi/2, \pi/2]$, we can write

$$
\begin{aligned}
d_0 &= r\cos\pi(\phi_{M-1})\,\cos\pi(\phi_{M-2})\cdots\cos\pi(\phi_2)\cos(\phi_1),\\
d_1 &= r\cos\pi(\phi_{M-1})\,\cos\pi(\phi_{M-2})\cdots\cos\pi(\phi_2)\sin(\phi_1),\\
d_2 &= r\cos\pi(\phi_{M-1})\,\cos\pi(\phi_{M-2})\cdots\cos\pi(\phi_3)\sin\pi(\phi_2),\\
&\vdots\\
d_{M-3} &= r\cos\pi(\phi_{M-1})\,\cos\pi(\phi_{M-2})\sin\pi(\phi_{M-3}),\\
d_{M-2} &= r\cos\pi(\phi_{M-1})\sin\pi(\phi_{M-2}),\\
d_{M-1} &= r\sin\pi(\phi_{M-1}),
\end{aligned}
\tag{42}
$$

$$
\begin{aligned}
\phi_1 &= \arctan\left(\frac{d_1}{d_0}\right), \quad \phi_2 = \arctan\left(\frac{d_2}{\sqrt{d_0^2 + d_1^2}}\right), \cdots,\\
\phi_{M-1} &= \arctan\left(\frac{d_{M-1}}{\sqrt{d_0^2 + d_1^2 + \cdots + d_{M-2}^2}}\right),\\
r &= \sqrt{d_0^2 + d_1^2 + \cdots + d_{M-2}^2 + d_{M-1}^2},
\end{aligned}
\tag{43}
$$

and thus, we define $M$D hypercomplex number $g$ and its conjugate $\bar{g}$ as

$$
g = d_0 + j_1 d_1 + j_2 d_2 + \cdots + j_{M-2}d_{M-2} + j_{M-1}d_{M-1},
\tag{44}
$$

$$
\bar{g} = d_0 - j_1 d_1 - j_2 d_2 - \cdots - j_{M-2}d_{M-2} - j_{M-1}d_{M-1}.
\tag{45}
$$

These hypercomplex numbers can be written in $M$-tuple representations as

$$
g = \begin{bmatrix} d_0 \\ d_1 \\ d_2 \\ d_3 \\ \vdots \\ d_{M-1} \end{bmatrix} \simeq \begin{bmatrix} r \\ \phi_1 \\ \phi_2 \\ \phi_3 \\ \vdots \\ \phi_{M-1} \end{bmatrix}, \text{ and } \bar{g} = \begin{bmatrix} d_0 \\ -d_1 \\ -d_2 \\ -d_3 \\ \vdots \\ -d_{M-1} \end{bmatrix} \simeq \begin{bmatrix} r \\ -\phi_1 \\ -\phi_2 \\ -\phi_3 \\ \vdots \\ -\phi_{M-1} \end{bmatrix}, \text{ and thus}
$$

$g\bar{g} = \|g\|^2 = r^2 \Rightarrow \|g\| = r$. The addive inverse of $g$ is $-g = \begin{bmatrix} -d_0 \\ -d_1 \\ -d_2 \\ -d_3 \\ \vdots \\ -d_{M-1} \end{bmatrix} \simeq \begin{bmatrix} r \\ \phi_1 + \pi \\ -\phi_2 \\ -\phi_3 \\ \vdots \\ -\phi_{M-1} \end{bmatrix}$, which

is equivalent to $(-1)g$ and then taking conjugation with respect to $j_2, j_3, \cdots, j_{M-1}$, i.e., $-g = (-1)\bar{g}_{j_2, j_3, \cdots j_{M-1}}$ where $\bar{g}_{j_2, j_3, \cdots j_{M-1}} = d_0 + j_1 d_1 - j_2 d_2 - \cdots - j_{M-2}d_{M-2} - j_{M-1}d_{M-1}$. The multipli-

cative inverse of $g$ is $1/g = \frac{\bar{g}}{g\bar{g}} = \frac{1}{r^2}\begin{bmatrix} d_0 \\ -d_1 \\ -d_2 \\ -d_3 \\ \vdots \\ -d_{M-1} \end{bmatrix} \simeq \begin{bmatrix} 1/r \\ -\phi_1 \\ -\phi_2 \\ -\phi_3 \\ \vdots \\ -\phi_{M-1} \end{bmatrix}$. The $M$-tuple representations in

SCS for basis $\{1, j_1, j_2, \cdots, j_{M-1}\}$, additive identity 0 and −1 are

$$
1 = \begin{bmatrix} 1 \\ 0 \\ 0 \\ 0 \\ \vdots \\ 0 \end{bmatrix}, j_1 = \begin{bmatrix} 1 \\ \pi/2 \\ 0 \\ 0 \\ \vdots \\ 0 \end{bmatrix}, j_2 = \begin{bmatrix} 1 \\ \phi_1 \\ \pi/2 \\ 0 \\ \vdots \\ 0 \end{bmatrix}, j_3 = \begin{bmatrix} 1 \\ \phi_1 \\ \phi_2 \\ \pi/2 \\ \vdots \\ 0 \end{bmatrix}, \cdots, j_{M-1} = \begin{bmatrix} 1 \\ \phi_1 \\ \phi_2 \\ \phi_3 \\ \vdots \\ \pi/2 \end{bmatrix},
$$

(46)

$$
0 = \begin{bmatrix} 0 \\ 0 \\ 0 \\ 0 \\ \vdots \\ 0 \end{bmatrix}, \ -1 = \begin{bmatrix} 1 \\ \pi \\ 0 \\ 0 \\ \vdots \\ 0 \end{bmatrix}.
$$

It is evident from (43) and (46) that the complex number $j_2$ exhibits an infinite set of representations as $\phi_1$ can assume any value. Similarly, $j_3$ manifests an infinite representations as both $\phi_1$ and $\phi_2$ can independently assume arbitrary values. Furthermore, the variable $j_{M-1}$ is characterized by an infinite set of representations, contingent upon the unconstrained values assumed by the variables $\phi_1, \phi_2, \ldots, \phi_{M-2}$. Thus, $j_{m-1}$ has $(m-2)$ degrees of freedom for $m \in \{3, 4, \cdots, M\}$.

To elucidate the unique characterization of the complex units within the set $\{j_1, j_2, j_3, \ldots, j_{M-1}\}$, constrained by the condition $j_1^2 = j_2^2 = j_3^2 = \cdots = j_{M-1}^2 = -1$, distinct angular values are introduced. Specifically, an angular assignment of $\phi_1 = \pi/2$ is prescribed, while the angular values $\phi_2, \phi_3, \ldots, \phi_{M-2}$ may assume either 0 or $\pi/2$. These angular assignments are established within the framework of the identity encapsulated in (46). It is noteworthy to mention that, unless explicitly specified otherwise, these angular values shall remain consistent throughout the subsequent analysis.

To obtain the generalized multiplication of two numbers $g_1$ and $g_2$, we write them in SCS $M$-tuple notations as

$$
g_1 = \begin{bmatrix} d_{01} \\ d_{11} \\ d_{21} \\ \vdots \\ d_{(M-1)1} \end{bmatrix} \simeq \begin{bmatrix} r_1 \\ \phi_{11} \\ \phi_{21} \\ \vdots \\ \phi_{(M-1)1} \end{bmatrix}, g_2 = \begin{bmatrix} d_{02} \\ d_{12} \\ d_{22} \\ \vdots \\ d_{(M-1)2} \end{bmatrix} \simeq \begin{bmatrix} r_2 \\ \phi_{12} \\ \phi_{22} \\ \vdots \\ \phi_{(M-1)2} \end{bmatrix}
$$

(47)

$$
\text{and } g_3 = \begin{bmatrix} d_{03} \\ d_{13} \\ d_{23} \\ \vdots \\ d_{(M-1)3} \end{bmatrix} \simeq \begin{bmatrix} r_3 \\ \phi_{13} \\ \phi_{23} \\ \vdots \\ \phi_{(M-1)3} \end{bmatrix}
$$

and, hereby, define the addition and spherical multiplication (SM) as

$$
g_1 + g_2 = \begin{bmatrix} d_{01} + d_{02} \\ d_{11} + d_{12} \\ d_{21} + d_{22} \\ \vdots \\ d_{(M-1)1} + d_{(M-1)2} \end{bmatrix} \text{ and } g_1 g_2 = \begin{bmatrix} r_1 r_2 \\ \phi_{11} + \phi_{12} \\ \phi_{21} + \phi_{22} \\ \vdots \\ \phi_{(M-1)1} + \phi_{(M-1)2} \end{bmatrix}, \tag{48}
$$

respectively. Moreover, in general $g_1(g_2 + g_3) \neq g_1 g_2 + g_1 g_3$, however, if $g_1 \in \mathbb{S}^2$, then $g_1(g_2 + g_3) = g_1 g_2 + g_1 g_3$. Using the proposed multiplication (48), we can decompose $g$ as

$$
g = \begin{bmatrix} r \\ \phi_1 \\ \phi_2 \\ \vdots \\ \phi_{M-1} \end{bmatrix} = \begin{bmatrix} r \\ 0 \\ 0 \\ 0 \\ \vdots \\ 0 \end{bmatrix} \begin{bmatrix} 1 \\ \phi_1 \\ 0 \\ 0 \\ \vdots \\ 0 \end{bmatrix} \begin{bmatrix} 1 \\ 0 \\ \phi_2 \\ 0 \\ \vdots \\ 0 \end{bmatrix} \cdots \begin{bmatrix} 1 \\ 0 \\ 0 \\ 0 \\ \vdots \\ \phi_{M-1} \end{bmatrix}, \tag{49}
$$

and thus, similar to 3D case (37) and (40), we obtain

$$
g = r \exp\left(\sum_{m=1}^{M-1} j_m \phi_m\right) \Rightarrow \ln g = \ln r + j_1 \phi_1 + \cdots + j_{M-1}\phi_{M-1}. \tag{50}
$$

There is complete uniqueness in the representation of $M$D numbers, akin to the 3D case, if we restrict $\phi_2, \phi_3, \cdots, \phi_{M-1} \in (-\pi/2, \pi/2)$, thereby excluding only two endpoints from the interval $[-\pi/2, \pi/2]$. This restriction can always be practically implemented by ensuring that $d_0$ and $d_1$ are not simultaneously zero in (44). Hence, it can be easily determined that the envisaged $M$D hypercomplex number system satisfies all the axioms prescribed in 3D Theorem 1. Consequently, we infer that the $M$D hypercomplex number system exemplifies a legitimate $M$-dimensional non-distributive normed division algebra.

## 5 A generalized vector space

A *vector space*, also known as a *linear space*, is a mathematical structure $V$ consisting of a set of elements as vectors, along with two operations, *vector addition*, and *scalar multiplication*, that satisfy specific properties. Formally, a vector space is defined as follows:

**Definition**. Let $V$ be a set of vectors, and let $F$ be a field ($\mathbb{R}$ or $\mathbb{C}$). A vector space over $F$ is a pair $(V, F)$ equipped with two operations:

- **Vector Addition**: A binary operation that assigns to each pair of vectors $u, v \in V$ a vector $u + v \in V$, such that the following properties hold for all vectors $u, v, w \in V$:

  1. Commutativity: $u + v = v + u$

  2. Associativity: $(u + v) + w = u + (v + w)$

  3. Identity Element: There exists a vector $\mathbf{0} \in V$, called the zero vector, such that $u + \mathbf{0} = u$ for all $u \in V$

4. Inverse Element: For each vector $u \in V$, there exists a vector $-u \in V$, called the additive inverse of $u$, such that $u + (-u) = \mathbf{0}$.

- **Scalar Multiplication**: A binary operation that assigns to each scalar $\alpha \in F$ and each vector $u \in V$ a vector $\alpha u \in V$, such that the following properties hold for all scalars $\alpha, \beta \in F$ and all vectors $u, v \in V$:

  1. Compatibility with field multiplication: $\alpha(\beta u) = (\alpha \beta)u$

  2. Identity element of the field: $1u = u$, where 1 is the multiplicative identity in the field $F$

  3. Distributivity over vector addition: $\alpha(u + v) = \alpha u + \alpha v$

  4. Distributivity over field addition: $(\alpha + \beta)u = \alpha u + \beta u$

  5. Zero scalar multiplication: $0u = \mathbf{0}$

  6. Scalar multiplication of the zero vector: $\alpha \mathbf{0} = \mathbf{0}$.

These properties collectively ensure that a vector space behaves well under vector addition and scalar multiplication. Vector spaces are fundamental in various mathematics, physics, and engineering branches, providing a general framework for studying linear phenomena.

Using the spherical multiplication (48) defined for the $MD$ hypercomplex number system, presented in Section 4, and motivated by the need to explore geometric and algebraic structures beyond standard Euclidean spaces, we introduce a *generalized real vector space*. This structure, $(V, +, \cdot, \times)$ with $V = \mathbb{R}^M$, contains a set of elements as vectors and supports three fundamental operations: *vector addition*, *scalar multiplication* and *vector multiplication*. Importantly, these operations conform to the established axioms of a *vector space* and additionally satisfy the following properties:

- **Vector Multiplication**: A binary operation that assigns to each pair of vectors $u, v \in V$ a vector $u \times v$, such that the following properties hold for all vectors $u, v, w \in V$:

  1. Closure: $u \times v \in V$

  2. Commutativity: $u \times v = v \times u$

  3. Associativity: $(u \times v) \times w = u \times (v \times w)$

  4. Identity Element: There exists a vector $1 \in V$ such that $u \times 1 = u$ for all $u \in V$.

  5. Inverse Element: For each nonzero vector $u \in V$, there exists a vector $u^{-1} \in V$, called the multiplicative inverse of $u$, such that $u \times u^{-1} = 1$

  6. Preservation of Norm: $\|u \times v\| = \|v\| \times \|u\|$.

Thus, analogously to how a real vector space equipped with vector addition $(V, +)$ exhibits the properties of an Abelian group, it is significant to recognize that a real vector space under the proposed vector multiplication $(V, \times)$ also forms an Abelian group. Thus, a non-distributive vector algebra is obtained. This observation emphasizes the intrinsic structural and algebraic properties inherent to vector addition and vector multiplication operations within the context of the proposed generalized real vector space. In this framework, scalars can be treated as vectors. However, vector multiplication corresponds to scalar multiplication only when the scalar $\alpha$ is a non-negative real number, i.e. $\alpha \geq 0$.

The vector space $\mathbb{C}^n$ demonstrates the isomorphism with $\mathbb{R}^{2n}$. This is established by a bijective mapping that leverages the well-known correspondence between complex numbers ($a +$

*ib*) and ordered pairs of real numbers $\begin{bmatrix} a \\ b \end{bmatrix}$. This mapping generalizes to complex vectors, where each element in $\mathbb{C}^n$ is mapped to a vector in $\mathbb{R}^{2n}$ by pairing its real and imaginary parts. The formally defined isomorphism ensures that the fundamental algebraic structure of vector addition and scalar multiplication is preserved. Consequently, from the perspective of their vector space structures, $\mathbb{C}^n$ and $\mathbb{R}^{2n}$ are isomorphic. Therefore, the above proposed generalized real vector space $\mathbb{R}^M$ can be easily extended to obtain a *generalized complex vector space* $\mathbb{C}^M$.

Introducing vector multiplication within a vector space framework paves the way for novel mathematical and physical investigations, potentially fueling theoretical and applied advancements in fields reliant on multidimensional analysis. This generalized model augments the fundamental mathematical structures crucial for comprehending complex phenomena in higher dimensions, offering a versatile tool for both theoretical and applied research endeavors. Extending the established vector space theory ushers in the potential for further mathematical innovation, enabling the exploration of novel algebraic and geometric configurations that could unlock more profound insights into multidimensional spaces. In essence, the proposed generalized real vector space signifies a notable stride forward in the study of algebraic structures, promising to enrich the mathematical structure with new theories and methodologies for delving into the complexities of multidimensional spaces.

## 6 Spherical multiplication as an invertible nonlinear map

To define spherical multiplication as an invertible nonlinear map, we specify both the forward operation and its inverse. In the context of the proposed spherical multiplication framework, we ensure that the transformation can be reversed, allowing recovery of the original vectors from the product vector.

**Definition**. Invertible nonlinear map: Let $\mathbf{u}$ and $\mathbf{v}$ be vectors in $\mathbb{R}^n$, represented in spherical coordinates as

$\mathbf{u} = [r_1, \phi_{11}, \phi_{21}, \ldots, \phi_{n1}]^T$ and $\mathbf{v} = [r_2, \phi_{12}, \phi_{22}, \ldots, \phi_{n2}]^T$. Define the spherical multiplication $\mathcal{T} : \mathbb{R}^n \times \mathbb{R}^n \to \mathbb{R}^n$ as:

$$\mathcal{T}(\mathbf{u}, \mathbf{v}) = [r_1 r_2, \phi_{11} + \phi_{12}, \phi_{21} + \phi_{22}, \cdots, \phi_{n1} + \phi_{n2}]^T.$$

To define the inverse operation, let $\mathbf{w} = \mathcal{T}(\mathbf{u}, \mathbf{v}) = [r, \phi_1, \phi_2, \ldots, \phi_n]^T$, where

$$r = r_1 r_2, \ \phi_1 = (\phi_{11} + \phi_{12}), \ \phi_2 = (\phi_{21} + \phi_{22}), \ \cdots, \ \phi_n = (\phi_{n1} + \phi_{n2}).$$

Assuming $\mathbf{u}$ is known and we want to recover $\mathbf{v}$, we can compute the components of $\mathbf{v}$ as

$$r_2 = \frac{r}{r_1}, \ \phi_{12} = (\phi_1 - \phi_{11}), \ \cdots, \ \phi_{n2} = (\phi_n - \phi_{n1}).$$

Similarly, if $\mathbf{v}$ is known and we want to recover $\mathbf{u}$, we compute

$$r_1 = \frac{r}{r_2}, \ \phi_{11} = (\phi_1 - \phi_{12}), \ \cdots, \ \phi_{n1} = (\phi_n - \phi_{n2}).$$

The properties of the invertible nonlinear map are as follows:

1. Non-distributive: The spherical multiplication remains non-distributive as

$$\mathcal{T}(\mathbf{u}, \mathbf{v} + \mathbf{w}) \neq \mathcal{T}(\mathbf{u}, \mathbf{v}) + \mathcal{T}(\mathbf{u}, \mathbf{w}),$$

which differentiates it from typical linear transformations.

2. Commutative: The multiplication is commutative, allowing the interchange of vectors as

$$\mathcal{T}(\mathbf{u}, \mathbf{v}) = \mathcal{T}(\mathbf{v}, \mathbf{u}).$$

3. Associative: The operation is associative as

$$\mathcal{T}(\mathcal{T}(\mathbf{u}, \mathbf{v}), \mathbf{w}) = \mathcal{T}(\mathbf{u}, \mathcal{T}(\mathbf{v}, \mathbf{w})).$$

4. Invertibility: Given the result of the multiplication and one of the original vectors, the other vector can be uniquely determined. This property makes the map invertible, ensuring the preservation of information and enabling the recovery of the initial vectors.

Some potential applications of the invertible nonlinear map are as follows: (i) Quantum Computing: Invertibility is crucial in quantum computing for reversible computations. The spherical multiplication could model invertible nonlinear quantum gates. (ii) Cryptography: Invertible nonlinear transformations are valuable in cryptographic algorithms, ensuring that encrypted data can be decrypted. (iii) Data Encoding: Invertible maps allow for data compression and encoding schemes where the original data can be perfectly reconstructed from the encoded data. (iv) Simulations and Modeling: In fields like physics and engineering, invertible nonlinear maps can simulate reversible processes, such as conservative force fields or reversible thermodynamic processes.

Thus, spherical multiplication as an invertible nonlinear map provides a robust mathematical framework that ensures information preservation and reversibility. Its properties and potential applications make it a powerful tool in various scientific and technological domains, offering unique advantages in modeling, simulation, and data processing.

# 7 Relationship between addition and multiplication through logarithms

The relationship between addition and multiplication can be understood through logarithms. By converting multiplication into addition in the logarithmic domain, we can simplify many mathematical problems and gain deeper insight into the nature of these operations. This is why logarithms are a powerful tool in mathematics and its applications.

Next, we show that the natural logarithms of hypercomplex numbers form an abelian group under addition because they satisfy all the necessary properties.

1. **Definition of the set**: The natural logarithm of a hypercomplex number $z = re^{i\phi}e^{j\theta}$ is:

$$\ln(z) = \ln(r) + i(\phi + 2k\pi) + j(\theta + \ell\pi),$$

where $r > 0$, $k, \ell \in \mathbb{Z}$, and the arguments can be any real number, however, we consider within the principal range $\phi \in (-\pi, \pi]$ and $\theta \in [-\pi/2, \pi/2]$ using (i) $\theta \mapsto \theta \bmod 2\pi$, and (ii) $\theta \mapsto \theta \,\hat{\mathrm{m}}\mathrm{od}\, \pi$ as defined in (26) that maps $\theta$ in the desired range $[-\pi/2, \pi/2]$.

- The set is defined as:

$$\{\ln(z) \mid z \in \mathbb{S}^3, z \neq 0\} = \{\ln(r) + i(\phi + 2k\pi) + j(\theta + \ell\pi) \mid r > 0, \ \phi, \theta \in \mathbb{R}, \ k, \ell \in \mathbb{Z}\}.$$

2. **Definition of the operation**: The operation considered is addition:

$$\ln(z_1) + \ln(z_2).$$

3. **Group properties**:

(a) **Closure**: For $z_1, z_2 \in \mathbb{S}^3 \setminus \{0\}$, let

$$\ln(z_1) = \ln(r_1) + i(\phi_1 + 2k_1\pi) + j(\theta_1 + \ell_1\pi),$$

and

$$\ln(z_2) = \ln(r_2) + i(\phi_2 + 2k_2\pi) + j(\theta_2 + \ell_2\pi),$$

then

$$\ln(z_1) + \ln(z_2) = \ln(r_1) + \ln(r_2) + i[(\phi_1 + \phi_2) + 2(k_1 + k_2)\pi] + j[(\theta_1 + \theta_2) + (\ell_1 + \ell_2)\pi].$$

Since $\ln(r_1 r_2) + i[(\phi_1 + \phi_2) + 2(k_1 + k_2)\pi] + j[(\theta_1 + \theta_2) + (\ell_1 + \ell_2)\pi]$ represents a valid hyper-complex logarithm, the set is closed under addition.

(b) **Associativity**: Addition of hypercomplex numbers is associative. Thus, for $\ln(z_1)$, $\ln(z_2)$, $\ln(z_3)$:

$$(\ln(z_1) + \ln(z_2)) + \ln(z_3) = \ln(z_1) + (\ln(z_2) + \ln(z_3)).$$

(c) **Identity Element**: The identity element for addition is 0. The logarithm of 1 is:

$$\ln(1) = 0.$$

(d) **Inverse Element**: For $\ln(z) = \ln(r) + i(\phi + 2k\pi) + j(\theta + \ell\pi)$, the inverse element is:

$$-\ln(z) = -\ln(r) - i(\phi + 2k\pi) - j(\theta + \ell\pi),$$

which is also in the set of natural logarithms of hypercomplex numbers.

(e) **Commutativity**: Addition of hypercomplex numbers is commutative. Therefore, for $\ln(z_1)$, $\ln(z_2)$:

$$\ln(z_1) + \ln(z_2) = \ln(z_2) + \ln(z_1).$$

The set of natural logarithms of hypercomplex numbers, under addition, satisfies the properties of closure, associativity, identity element, inverse element, and commutativity. Therefore, it forms an abelian group under addition.

## 8 Numerical simulation results

In this section, we present some numerical simulation examples as follows.

### 8.1 Illustrative simulation: Bloch sphere

The Bloch sphere, named after the physicist Felix Bloch, is a fundamental concept in quantum mechanics, particularly in quantum computing and quantum information theory. It provides

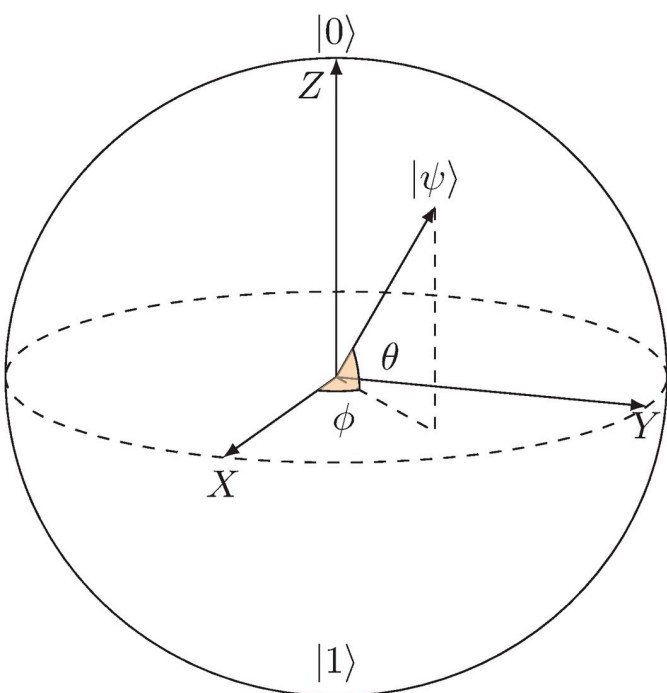

**Fig 7. The Bloch sphere representation, where the azimuth angle $\phi \in [0, 2\pi)$ is measured from the positive $X$ axis and the elevation angle $\theta \in [-\pi/2, \pi/2]$ is measured from the $XY$ plane.** The positive $Z$ axis corresponds to the state $|0\rangle$, the negative $Z$ axis corresponds to the state $|1\rangle$, and any point on the Bloch sphere can be uniquely represented by a pair of angles $(\phi, \theta)$.

a geometric representation that visually depicts the quantum states of a two-level quantum system, commonly known as a qubit. The Bloch sphere is a unit sphere where the poles represent the classical states $|0\rangle = \begin{bmatrix} 1 \\ 0 \end{bmatrix}$ and $|1\rangle = \begin{bmatrix} 0 \\ 1 \end{bmatrix}$, and any point on the surface corresponds to a unique quantum state. The azimuthal and polar angles of a point on the Bloch sphere encode the probability amplitudes of the quantum state in the computational basis. This representation is invaluable for illustrating quantum operations, visualizing quantum gates, and understanding the dynamics of quantum systems. The Bloch sphere is a powerful tool for both pedagogical purposes and advanced research in quantum computing and quantum information science.

In the literature, the polar angle $\theta \in [0, \pi]$ is measured from the positive $Z$-axis. However, in our representation $\theta$ is the elevation angle measured from the $XY$ plane as shown in Fig 7. The poles of the Bloch sphere typically represent the basis states of the qubit, often denoted as $|0\rangle$ and $|1\rangle$. A quantum state $|\psi\rangle$ can be represented as a superposition of pure states $|0\rangle$ and $|1\rangle$ as

$$|\psi\rangle = \cos\pi\left(\left(\frac{\pi}{2} - \theta\right)/2\right)|0\rangle + \sin\pi\left(\left(\frac{\pi}{2} - \theta\right)/2\right)e^{i\phi}|1\rangle, \quad \phi \in [0, 2\pi), \quad \theta \in [-\pi/2, \pi/2]. \tag{51}$$

Thus, the quantum state $|\psi\rangle$ is uniquely mapped to a point in the Bloch sphere that can be represented in SCS as

$$|\psi\rangle = \begin{bmatrix} \cos\pi\left(\left(\frac{\pi}{2} - \theta\right)/2\right) \\ \sin\pi\left(\left(\frac{\pi}{2} - \theta\right)/2\right)e^{i\phi} \end{bmatrix} \simeq \begin{bmatrix} 1 \\ \phi \\ \theta \end{bmatrix}. \tag{52}$$

Therefore, using (52), we can write quantum states in the Bloch sphere as

$$|0\rangle = \begin{bmatrix} 1 \\ 0 \end{bmatrix} \simeq \begin{bmatrix} 1 \\ 0 \\ \pi/2 \end{bmatrix}, \qquad |1\rangle = \begin{bmatrix} 0 \\ 1 \end{bmatrix} \simeq \begin{bmatrix} 1 \\ 0 \\ -\pi/2 \end{bmatrix}, \tag{53}$$

$$|+\rangle = \frac{1}{\sqrt{2}}(|0\rangle + |1\rangle) = \frac{1}{\sqrt{2}} \begin{bmatrix} 1 \\ 1 \end{bmatrix} \simeq \begin{bmatrix} 1 \\ 0 \\ 0 \end{bmatrix}, \; |-\rangle = \frac{1}{\sqrt{2}}(|0\rangle - |1\rangle) = \frac{1}{\sqrt{2}} \begin{bmatrix} 1 \\ -1 \end{bmatrix} \simeq \begin{bmatrix} 1 \\ \pi \\ 0 \end{bmatrix}, \tag{54}$$

$$|i\rangle = \frac{1}{\sqrt{2}}(|0\rangle + i|1\rangle) = \frac{1}{\sqrt{2}} \begin{bmatrix} 1 \\ i \end{bmatrix} \simeq \begin{bmatrix} 1 \\ \pi/2 \\ 0 \end{bmatrix}, \; |-i\rangle = \frac{1}{\sqrt{2}}(|0\rangle - i|1\rangle) = \frac{1}{\sqrt{2}} \begin{bmatrix} 1 \\ -i \end{bmatrix} \simeq \begin{bmatrix} 1 \\ -\pi/2 \\ 0 \end{bmatrix}, \tag{55}$$

where set $\{|0\rangle, |1\rangle\}$, $\{|+\rangle, |-\rangle\}$, and $\{|i\rangle, |-i\rangle\}$ represent the $Z$, $X$, and $Y$ bases, respectively. Thus, for example, using the proposed multiplication (16) and division (17), we obtain and define valid multiplication of quantum states and their inverses as

$$|0\rangle \times |1\rangle \simeq \begin{bmatrix} 1 \\ 0 \\ \pi/2 \end{bmatrix} \times \begin{bmatrix} 1 \\ 0 \\ -\pi/2 \end{bmatrix} = \begin{bmatrix} 1 \\ 0 \\ 0 \end{bmatrix} \simeq |+\rangle \text{ and } (|0\rangle)^{-1} = |1\rangle, \tag{56}$$

$$|+\rangle \times |-\rangle \simeq \begin{bmatrix} 1 \\ 0 \\ 0 \end{bmatrix} \times \begin{bmatrix} 1 \\ \pi \\ 0 \end{bmatrix} = \begin{bmatrix} 1 \\ \pi \\ 0 \end{bmatrix} \simeq |-\rangle, \quad (|+\rangle)^{-1} = |+\rangle \text{ and } (|-\rangle)^{-1} = |-\rangle, \tag{57}$$

$$|i\rangle \times |-i\rangle \simeq \begin{bmatrix} 1 \\ \pi/2 \\ 0 \end{bmatrix} \times \begin{bmatrix} 1 \\ -\pi/2 \\ 0 \end{bmatrix} = \begin{bmatrix} 1 \\ 0 \\ 0 \end{bmatrix} \simeq |+\rangle \text{ and } (|i\rangle)^{-1} = |-i\rangle. \tag{58}$$

From (51) and (52), we observe that the designated pair of quantum states, identified by antipodal locations on the Bloch sphere, forms a basis. This basis can be expressed as

$$\{\psi_1, \psi_2\} = \left\{ \begin{bmatrix} \cos\pi\left(\left(\frac{\pi}{2} - \theta\right)/2\right) \\ \sin\pi\left(\left(\frac{\pi}{2} - \theta\right)/2\right)e^{i\phi} \end{bmatrix}, \begin{bmatrix} \cos\pi\left(\left(\frac{\pi}{2} + \theta\right)/2\right) \\ \sin\pi\left(\left(\frac{\pi}{2} + \theta\right)/2\right)e^{i(\phi+\pi)} \end{bmatrix} \right\} \simeq \left\{ \begin{bmatrix} 1 \\ \phi \\ \theta \end{bmatrix}, \begin{bmatrix} 1 \\ \phi + \pi \\ -\theta \end{bmatrix} \right\}, \tag{59}$$

where the defining characteristic of these states lies in their distinct angular coordinates $\phi$ and

$\theta$. Notably, the well-established $X$, $Y$, and $Z$ bases manifest as specific cases within this framework. Moreover, the global phase of quantum states lacks physical significance and observable consequences. Mathematically, if $|\psi\rangle$ represents a valid quantum state, then any state of the form $e^{i\phi}|\psi\rangle$ is considered physically equivalent to $|\psi\rangle$. Consequently, states such as $\begin{bmatrix} 0 \\ 1 \end{bmatrix}$, $\begin{bmatrix} 0 \\ -1 \end{bmatrix}$, $\begin{bmatrix} 0 \\ i \end{bmatrix}$ and $e^{i\phi}\begin{bmatrix} 0 \\ 1 \end{bmatrix}$ are deemed equivalent.

In quantum mechanics, any unit norm vector in the complex vector space $\mathbb{C}^2$ represents a valid quantum state for a qubit, a fundamental principle in quantum theory. A unit vector in its associated Hilbert space describes the state of a quantum system, and for a qubit, the general quantum state form is $|\psi\rangle = \alpha|0\rangle + \beta|1\rangle$. Here, $\alpha$ and $\beta$ are complex numbers, while $|0\rangle$ and $|1\rangle$ are basis states. The condition $\|\alpha\|^2 + \|\beta\|^2 = 1$ ensures normalization, making any unit norm vector in $\mathbb{C}^2$ represent a valid quantum state for a qubit. The principle of superposition permits the combination of quantum states, resulting in a valid quantum state as long as the resulting vector is normalized. This normalization guarantees that the probabilities of all possible outcomes sum to 1. Mathematically, given two quantum states $|\psi_1\rangle$ and $|\psi_2\rangle$, their superposition and normalization $|\psi\rangle = (\alpha|\psi_1\rangle + \beta|\psi_2\rangle)/\sqrt{\|\alpha\|^2 + \|\beta\|^2}$ remain valid for all $\alpha, \beta \in \mathbb{C}$ such that $\sqrt{\|\alpha\|^2 + \|\beta\|^2} \neq 0$. This consistency ensures that the probabilities associated with different measurement outcomes align with the principles of quantum mechanics.

We postulate that analogous to the principles of superposition and normalization, the proposed vector multiplication of two quantum states yields a valid quantum state. In mathematical terms, considering two given quantum states $|\psi_1\rangle$ and $|\psi_2\rangle$, their vector multiplication $|\psi_3\rangle = |\psi_1\rangle \times |\psi_2\rangle$ is deemed a valid quantum state. In contrast to vector addition scenarios, normalization is unnecessary, as vector multiplication preserves the norm. The physical interpretation of multiplication can be considered as (i) Evolution: the operation $|\psi_1\rangle \times |\psi_2\rangle$ can represent the transition of a state $|\psi_1\rangle$ under the influence or operator $|\psi_2\rangle$, resulting in a new state $|\psi_3\rangle$, akin to time evolution in quantum mechanics. (ii) Interaction: If $|\psi_1\rangle$ and $|\psi_2\rangle$ represent states of two interacting quantum systems, $|\psi_1\rangle \times |\psi_2\rangle$ represents the state resulting from their interaction, capturing the effects of entanglement or mutual influence. Traditional quantum mechanics relies on linear unitary evolution. Our nonlinear approach may offer new ways for state evolution, potentially uncovering novel insights or behaviors where linear approximations are insufficient.

There are a few physical interpretations that align with the nonlinear nature of the influence. We can imagine quantum states represented by vectors as magnetic moments or electric charges, with the influence vector representing an external magnetic or electric field, causing nonlinear changes in orientation similar to magnetic moments aligning with an external field. Quantum states could also represent the spin states of particles, where the influencing vector represents the spin of another particle, modeling interactions akin to spin-spin coupling with nonlinear dynamics. In quantum computation, the influencing vector can transform quantum states as qubits, simulating quantum gate operations with nonlinear characteristics that extend beyond standard unitary gates. Additionally, quantum states evolving in nonlinear potentials could be influenced by vectors representing aspects of the potential or interactions within the system, reflecting the nonlinear evolution relevant to fields like nonlinear optics or chaotic quantum systems. Finally, quantum states in open systems interacting with an environment, where the influencing vector represents environmental effects, can model the complex, nonlinear evolution of states under decoherence or dissipation using multiplication.

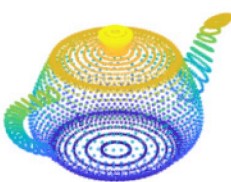
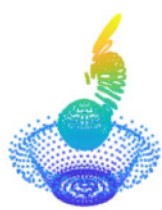

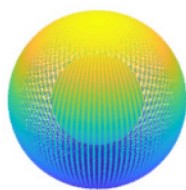
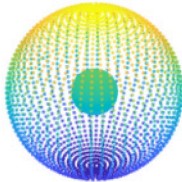

**Fig 8.** A point cloud image consist of a set of points in 3D coordinate system with $\{x_k, y_k, z_k\}_{k=1}^{N}$ geometric coordinates stored in $N \times 3$ matrix (i) Top-row left image (5184×3) and its squared image on right. (ii) Bottom-row left image (20402 × 3) represents two spheres of radius $r = 1$ (outer sphere) and $r = 1/2$ (inner sphere) and its squared image on right where outer sphere radius remain same, and inner sphere radius changed to 1/4.

Moreover, since the spherical multiplication forms an Abelian group, preserves the norm, and provides an invertible nonlinear map, it has unique properties for representing certain types of invertible nonlinear quantum gates in quantum systems. This method represents an alternative avenue for representing a quantum state as the multiplication of given states, thereby serving as a potential complement to the prevailing superposition paradigm, which is scalar multiplication and vector addition. It is noteworthy to mention that a non-distributive scator algebra has previously been employed to model the evolution and collapse of quantum wave functions [17].

## 8.2 Visual representation: Point cloud image 1

A point cloud image is composed of a collection of $N$ points within a three-dimensional (3D) coordinate system, where the geometric coordinates are designated as $x$, $y$, and $z$, and are stored in an $N \times 3$ matrix as $\{x_k, y_k, z_k\}_{k=1}^{N}$ which is equivalent to set $\{r_k, \phi_k, \theta_k\}_{k=1}^{N}$, and corresponding squared image is $\{r_k^2, 2\phi_k, 2\theta_k\}_{k=1}^{N}$. The considered example, depicted in Fig 8, encompasses the following elements: (i) the top-left image in the top row, with dimensions $5184 \times 3$, along with its corresponding squared image on the right; and (ii) the bottom-left image in the bottom row, measuring $20402 \times 3$, which represents two spheres with radii $r = 1$ (outer sphere) and $r = 1/2$ (inner sphere). The squared image that depicts this arrangement is presented on the right. It is noteworthy that, in this representation, the radius of the outer sphere remains constant, while the radius of the inner sphere is modified to 1/4.

## 8.3 Visual representation: Point cloud image 2

In this case, we delve into a set of point cloud images, specifically focusing on the top-row left image with dimensions $5184 \times 3$, i.e., $\{r_k, \phi_k, \theta_k\}_{k=1}^{N}$ with $N = 5184$. Subsequent images in the

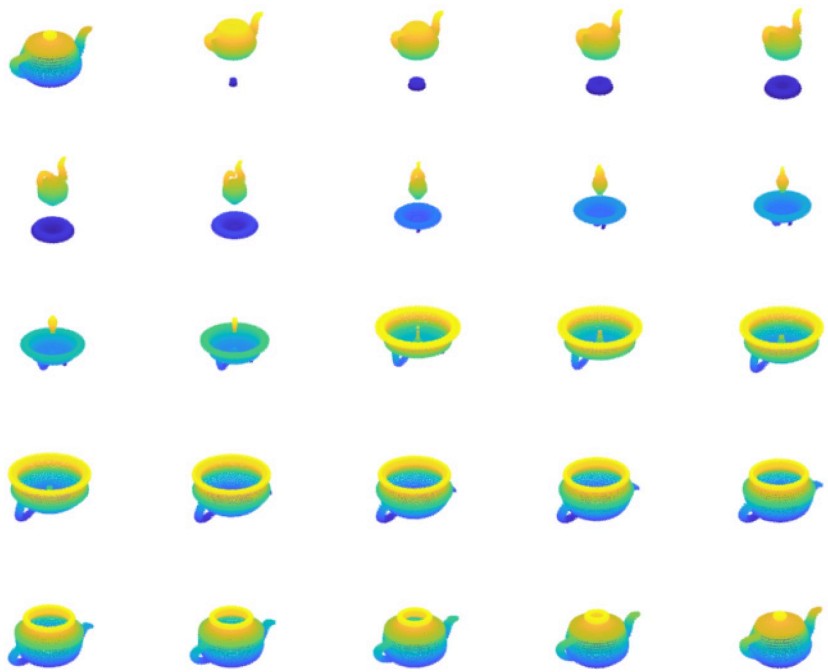

**Fig 9. A set of point cloud images: From the leftmost image (5184 × 3) of the top-row other images are obtained by clanging the phase angle θ ∈ [0, π] in step of π/24.** The right-most image of the bottom-row is the same as leftmost image of the top-row.

top row are generated by systematically varying the phase angle $\theta$ within the range $[0, \pi]$, with intervals set at $\pi/24$, using the proposed multiplication method, i.e., $\{r_k, \phi_k, \theta_k\}_{k=1}^{N} \times \{1, 0, \theta_\ell\}_{\ell=1}^{24}$ with $\theta_\ell = \ell \times \pi/24$. This phased transformation results in a sequence of point cloud images, each corresponding to a distinct value of $\theta$ within the specified range. Intriguingly, the right-most image of the the bottom row is identical to the left-most image of the top row, as visually represented in Fig 9. This recurrence signifies a specific condition or point in the phased evolution of the point cloud, demonstrating the cyclical nature of the proposed multiplication process. The detailed exploration of this progression allows for a comprehensive understanding of the impact of varying the phase angle on the resultant point cloud images.

## 8.4 Visual representation: Point cloud image 3

A collection of point cloud images representing a solid cube (132651 × 3) is depicted in Fig 10. Two distinct variations are explored:

(a) Commencing with the top-leftmost image in the top row, 24 additional images are generated by modulating the phase angle $\phi$ within the interval $\phi \in [0, 2\pi]$, with increments set at $2\pi/24$, i.e., $\{r_k, \phi_k, \theta_k\}_{k=1}^{N} \times \{1, \phi_\ell, 0\}_{\ell=1}^{24}$ with $\phi_\ell = \ell \times 2\pi/24$. In particular, the rightmost image in the bottom row mirrors the leftmost image in the top row. The variation in $\phi$ signifies the rotation around the $Z$-axis that does not alter the shape of a three-dimensional object.

(b) Alternatively, the top left image in the top row undergoes modification by altering the phase angle $\theta$ within the range $\theta \in [0, \pi]$, with increments of $\pi/24$, producing 24 distinct

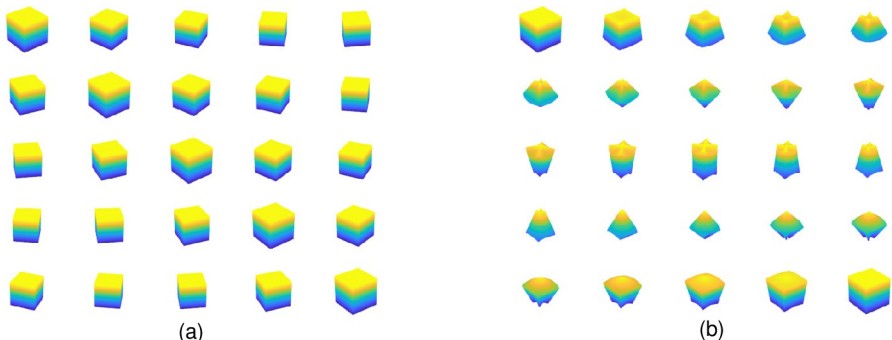

(a) (b)

**Fig 10.** A set of point cloud images of solid cube (132651 × 3): (a) From the leftmost image in the top row, other images are obtained by varying the phase angle $\phi$ within the range $\phi \in [0, 2\pi]$, with intervals set at $2\pi/24$. The rightmost image of the bottom row is identical to the leftmost image of the top row. (b) The leftmost image of the top-row is varied by changing the phase angle $\theta$ within the range $\theta \in [0, \pi]$ in increments of $\pi/24$ to produce other images. The right-most image of the bottom-row is identical to the left-most image of the top-row.

images. In this case as well, the bottom-rightmost image in the bottom row corresponds to the top-leftmost image in the top row.

## 8.5 Dynamic earth phenomena simulation

Our investigation yields a noteworthy observation, elucidating a correlation between the spherical coordinates system delineated by (3), as depicted in Fig 11, and dynamic Earth

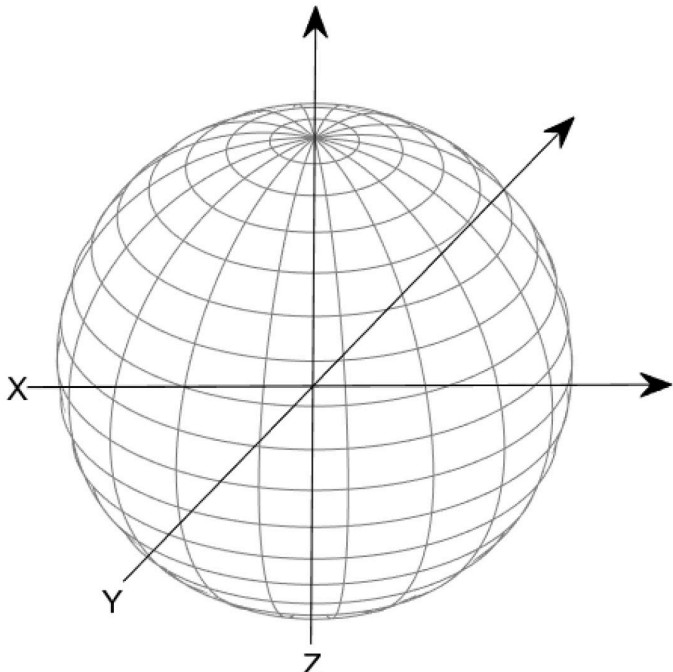

**Fig 11. Spherical coordinates system representation by (3) where continuous increase in azimuth angle $\phi$ with period $2\pi$ depicts spin of Earth around $Z$-axis (north and south pole), and continuous increase in elevation angle $\theta$ with period $\pi$ depicts flow of magnetic flux along longitudes with source at south pole ($\theta = -\pi/2$) and sink at north pole ($\theta = \pi/2$).**

phenomena such as its rotation around the $Z$-axis and the flux dynamics of the magnetic field. Within this framework, a continual increase in the azimuth angle $\phi$ over a $2\pi$ period faithfully emulates the rotational motion of Earth along its axis, distinguishing between the north and south poles. Currently, the uninterrupted evolution of the elevation angle $\theta$ over a $\pi$ period manifests the flow pattern of magnetic flux along longitudes. It is noteworthy to mention that in this representation the origin of the flux is at the south pole ($\theta = -\pi/2$) and its termination is at the north pole ($\theta = \pi/2$). This discerning analogy establishes a meaningful connection between the mathematical representation of spherical coordinates and the physical phenomena associated with the rotation of Earth and the flow of magnetic flux, thereby contributing to a nuanced understanding of their interrelation.

## 9 Conclusion

The fundamental and most important contribution of this study is the introduction of generalized hypercomplex numbers and the non-distributive normed division algebra in all dimensions. Notably, this framework seamlessly converges with the conventional theories of $\mathbb{R}$ and $\mathbb{C}$ spaces, shedding light on the geometric properties of vectors within these spaces. To ensure a broad applicability of this generalization, an innovative solution is developed, encompassing a non-distributive normed division algebra and a novel multiplication operation defined within the spherical coordinate system. Importantly, this new multiplication operation remains fully compatible with the established multiplication operation of numbers in $\mathbb{C}$. The proposed framework for generalized hypercomplex numbers, coupled with the inventive derivation approach, holds the potential to usher in a new era of higher-dimensional algebra. The anticipated applications could potentially span a spectrum of fields, including science, engineering, and technology, promising advancements that may prove indispensable in the near future.

**Appendix A. Proposed Generalized ($M$D) Hypercomplex Number System**: The construction of a 3-dimensional hypercomplex number system, $\mathbb{S}^3$, admits a natural extension to an $M$-dimensional hypercomplex number system, $\mathbb{S}^M$, via the application of the generalized $M$-dimensional spherical coordinate system. In this framework, the angular parameter $\theta$ is measured from the $Z$-axis. This approach deviates from the previous setting where $\theta$ is measured with respect to the $XY$ plane. As an illustrative example, the 3-dimensional hypercomplex number system can be explicitly formulated for all $\phi \in [0, 2\pi)$ and $\theta \in [0, \pi]$, where $\phi$ and $\theta$ represent the azimuthal and polar angles, respectively, as:

$$
\begin{aligned}
a &= r\,\text{sinpi}(\theta)\cos(\phi), \quad b = r\,\text{sinpi}(\theta)\sin(\phi), \quad c = r\,\text{cospi}(\theta), \\
r &= \sqrt{a^2 + b^2 + c^2}, \quad \phi = \arctan\left(\frac{b}{a}\right) \in [0, 2\pi), \quad \theta = \arctan\left(\frac{\sqrt{a^2 + b^2}}{c}\right) \in [0, \pi],
\end{aligned}
\tag{60}
$$

where the $\pi$-periodic functions $\text{cospi}(\theta) = \text{cospi}(\theta + n\pi)$ and $\text{sinpi}(\theta) = \text{sinpi}(\theta + n\pi) = |\sin(\theta)|$ for all $n \in \mathbb{Z}$ and $\theta \in \mathbb{R}$, are defined as

$$
\text{sinpi}(\theta) = \begin{cases}
\sin(\theta), & \text{for } \theta \in \bigcup_{n=-\infty}^{\infty} [2\pi n, \pi + 2\pi n], \\[2mm]
-\sin(\theta), & \text{for } \theta \in \bigcup_{n=-\infty}^{\infty} (\pi + 2\pi n, 2\pi + 2\pi n),
\end{cases}
\tag{61}
$$

$$\text{and} \quad \text{cospi}(\theta) = \begin{cases} \cos(\theta), & \text{for } \theta \in \bigcup_{n=-\infty}^{\infty} [2\pi n, \pi + 2\pi n], \\ -\cos(\theta), & \text{for } \theta \in \bigcup_{n=-\infty}^{\infty} (\pi + 2\pi n, 2\pi + 2\pi n), \end{cases} \tag{62}$$

where the intervals form a complete, non-overlapping partition of the real line as

$$\mathbb{R} = \left\{ \bigcup_{n=-\infty}^{\infty} [2n\pi, \pi + 2\pi n] \right\} \bigcup \left\{ \bigcup_{n=-\infty}^{\infty} (\pi + 2\pi n, 2\pi + 2\pi n) \right\}. \tag{63}$$

Therefore,

$$\text{sinpi}(\theta)/\cos\text{pi}(\theta) = \tan\text{pi}(\theta) = \tan(\theta). \tag{64}$$

In this representation $\theta \in \mathbb{R}$ with the following mappings

$$\theta \mapsto \theta \,\widetilde{\text{mod}}\, \pi = \begin{cases} \theta & \text{if } \theta \in [0, \pi], \\ \theta - m\pi & \text{if } \theta > \pi, \\ \theta + m\pi & \text{if } \theta < -\pi, \end{cases} \tag{65}$$

where $m$ is the smallest positive integer such that the result lies within $[0, \pi]$. Thus, we can write (1) in SCS as

$$g = r\,\text{sinpi}(\theta)\cos(\phi) + ir\,\text{sinpi}(\theta)\sin(\phi) + jr\,\text{cospi}(\theta), \tag{66}$$

where $g$ is $g(r, \phi, \theta)$, $\phi$ is an azimuth angle, and $\theta$ is an polar angle form $Z$-axis as shown in Fig 12. We can extend this and in general, with $\phi_1 \in [0, 2\pi)$ and $\phi_2, \phi_3, \cdots, \phi_{M-1} \in [0, \pi]$, we can write

$$\begin{aligned} d_0 &= r\,\text{sinpi}(\phi_{M-1})\,\text{sinpi}(\phi_{M-2}) \cdots \text{sinpi}(\phi_2)\cos(\phi_1), \\ d_1 &= r\,\text{sinpi}(\phi_{M-1})\,\text{sinpi}(\phi_{M-2}) \cdots \text{sinpi}(\phi_2)\sin(\phi_1), \\ d_2 &= r\,\text{sinpi}(\phi_{M-1})\,\text{sinpi}(\phi_{M-2}) \cdots \text{sinpi}(\phi_3)\,\text{cospi}(\phi_2), \\ &\vdots \\ d_{M-3} &= r\,\text{sinpi}(\phi_{M-1})\,\text{sinpi}(\phi_{M-2})\,\text{cospi}(\phi_{M-3}), \\ d_{M-2} &= r\,\text{sinpi}(\phi_{M-1})\,\text{cospi}(\phi_{M-2}), \\ d_{M-1} &= r\,\text{cospi}(\phi_{M-1}), \end{aligned} \tag{67}$$

$$\begin{aligned} \phi_1 &= \arctan\left(\frac{d_1}{d_0}\right), \quad \phi_2 = \arctan\left(\frac{\sqrt{d_0^2 + d_1^2}}{d_2}\right), \cdots, \\ \phi_{M-1} &= \arctan\left(\frac{\sqrt{d_0^2 + d_1^2 + \cdots + d_{M-2}^2}}{d_{M-1}}\right), \\ r &= \sqrt{d_0^2 + d_1^2 + \cdots + d_{M-2}^2 + d_{M-1}^2}. \end{aligned} \tag{68}$$

The operations of addition and spherical multiplication remain consistent with their definitions as specified in (47) and (48), respectively, within the framework of this methodology.

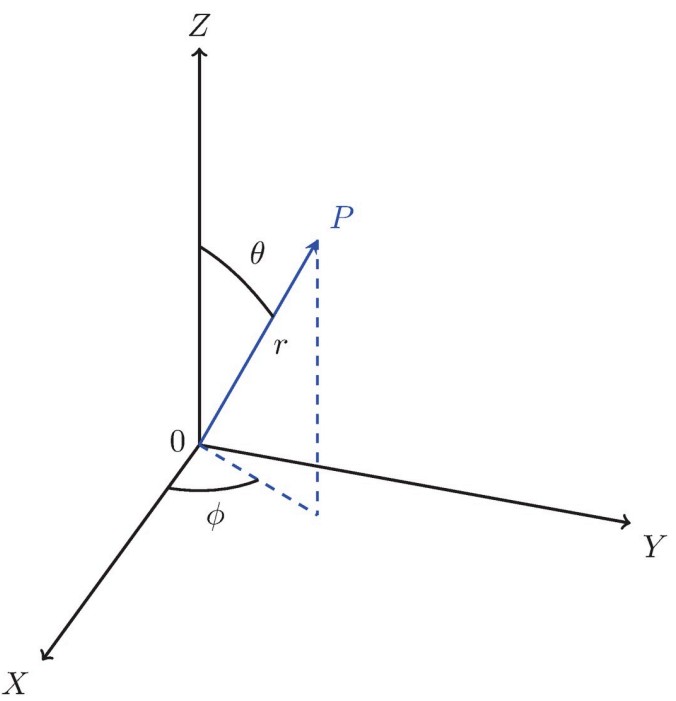

**Fig 12. A point $P$ in the considered spherical co-ordinate system, where radius ($r$), azimuth angle ($\phi$ rad), and polar angle ($\theta$ rad) measured from $Z$-axis are shown.**

However, the representation of the basis $\{1, j_1, \cdots, j_{M-1}\}$ is less intuitive and different from (46).

## Acknowledgments

The authors sincerely thank the anonymous reviewers for their valuable time, insightful feedback, and suggestions, which significantly improved this work. Special thanks to Prof. Govind S. Krishnaswami (Chennai Mathematical Institute) and Dr. Sonakshi Sachdev for their valuable input. We would also like to express our gratitude to Prisha Singh for posing insightful questions that initiated the journey of this work.

## Author Contributions

**Conceptualization:** Pushpendra Singh.

**Data curation:** Pushpendra Singh.

**Formal analysis:** Pushpendra Singh, Anubha Gupta, Shiv Dutt Joshi.

**Funding acquisition:** Anubha Gupta.

**Investigation:** Pushpendra Singh.

**Methodology:** Pushpendra Singh, Anubha Gupta, Shiv Dutt Joshi.

**Software:** Pushpendra Singh.

**Supervision:** Shiv Dutt Joshi.

**Validation:** Pushpendra Singh, Anubha Gupta, Shiv Dutt Joshi.

**Visualization:** Pushpendra Singh, Shiv Dutt Joshi.

**Writing – original draft:** Pushpendra Singh, Anubha Gupta.

**Writing – review & editing:** Pushpendra Singh, Anubha Gupta, Shiv Dutt Joshi.

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
