## [Decision Letter · Decision Letter 0]

30 Aug 2024

PONE-D-24-24879On the hypercomplex numbers and normed division algebra of all dimensions: A unified multiplicationPLOS ONE

Dear Dr. Singh,

Thank you for submitting your manuscript to PLOS ONE. After careful consideration, we feel that it has merit but does not fully meet PLOS ONE’s publication criteria as it currently stands. Therefore, we invite you to submit a revised version of the manuscript that addresses the points raised during the review process.

We look forward to receiving your revised manuscript.

Kind regards,

Sania Asif

Guest Editor

PLOS ONE

Journal Requirements:

When submitting your revision, we need you to address these additional requirements. 1. Please ensure that your manuscript meets PLOS ONE's style requirements, including those for file naming. The PLOS ONE style templates can be found at https://journals.plos.org/plosone/s/file?id=wjVg/PLOSOne_formatting_sample_main_body.pdf and https://journals.plos.org/plosone/s/file?id=ba62/PLOSOne_formatting_sample_title_authors_affiliations.pdf 2. Please update your submission to use the PLOS LaTeX template. The template and more information on our requirements for LaTeX submissions can be found at http://journals.plos.org/plosone/s/latex 3. We note that your Data Availability Statement is currently as follows: All relevant data are within the manuscript and its Supporting Information files. Please confirm at this time whether or not your submission contains all raw data required to replicate the results of your study. Authors must share the “minimal data set” for their submission. PLOS defines the minimal data set to consist of the data required to replicate all study findings reported in the article, as well as related metadata and methods (https://journals.plos.org/plosone/s/data-availability#loc-minimal-data-set-definition). For example, authors should submit the following data: - The values behind the means, standard deviations and other measures reported;- The values used to build graphs;- The points extracted from images for analysis. Authors do not need to submit their entire data set if only a portion of the data was used in the reported study. If your submission does not contain these data, please either upload them as Supporting Information files or deposit them to a stable, public repository and provide us with the relevant URLs, DOIs, or accession numbers. For a list of recommended repositories, please see https://journals.plos.org/plosone/s/recommended-repositories. If there are ethical or legal restrictions on sharing a de-identified data set, please explain them in detail (e.g., data contain potentially sensitive information, data are owned by a third-party organization, etc.) and who has imposed them (e.g., an ethics committee). Please also provide contact information for a data access committee, ethics committee, or other institutional body to which data requests may be sent. If data are owned by a third party, please indicate how others may request data access. 4. Please ensure that you refer to Figure 5 in your text as, if accepted, production will need this reference to link the reader to the figure.

**Additional Editor Comments:**

Manuscript ID: PONE-D-24-24879

Title: On the hypercomplex numbers and normed division algebra of all dimensions: A unified multiplication.

Journal: PLOS ONE

Dear Pushpendra Singh,

Thanks for submitting your manuscreipt to the Journal, PLOS ONE.

1. Please revise your paper according to the referee's advice in the attachment.

2. Moreover, please check the whole paper carefully again to avoid typos and mistakes.

Please do not hesitate to contact us if you have any questions regarding the

revision of your manuscript or if you need more time. We look forward to

hearing from you soon.

Many thanks!

Best regards

Sania Asif

Reviewers' comments:

Reviewer's Responses to Questions

**Comments to the Author**

1. Is the manuscript technically sound, and do the data support the conclusions?

Reviewer #1: Yes

Reviewer #2: Yes

2. Has the statistical analysis been performed appropriately and rigorously? 

Reviewer #1: N/A

Reviewer #2: Yes

3. Have the authors made all data underlying the findings in their manuscript fully available?

Reviewer #1: Yes

Reviewer #2: Yes

4. Is the manuscript presented in an intelligible fashion and written in standard English?

Reviewer #1: Yes

Reviewer #2: Yes

5. Review Comments to the Author

Reviewer #1: pdf file attached with detailed comments

pdf file attached with detailed comments

Reviewer #2: In this paper the authors present a comprehensive generalization of complex numbers and the Euler identity in higher dimensions, shedding light on the geometric properties of vectors within these extended spaces. Moreover, they elucidated the practical applications of the proposed methodology as a viable alternative for expressing a quantum state through the multiplication of specified quantum states, thereby offering a potential complement to the established superposition paradigm. Importantly, they explored its utility in point cloud image processing. The paper is well supported by the examples and figures which puts life to the paper. The paper is well written and fits to the scope of the journal. I recommend acceptance.

6. PLOS authors have the option to publish the peer review history of their article (what does this mean?). If published, this will include your full peer review and any attached files.

Reviewer #1: No

Reviewer #2: No

---

## [Author Response · Author response to Decision Letter 0]

18 Sep 2024

Dear Editor:

First of all, please accept our thanks for devoting time and coordinating the reviews of the submitted manuscript Ref. No. PONE-D-24-24879 titled ``On the hypercomplex numbers and normed division algebras in all dimensions: A unified multiplication'' for possible publication in PLOS ONE. We submitted the manuscript on June 19, 2024 and received the review comments from the reviewers on August 31, 2024. We appreciate the time and effort you and the anonymous reviewers devoted to reviewing and providing useful comments to help us improve the quality of the manuscript. 

We have done our best to address the comments of the reviewers. We, hereby, submit: (1) a response letter with detailed replies to each of the reviewers’ comments and (2) a revised manuscript that we believe to have significantly improved after incorporating suggestions of the reviewers. 

Please do thank the reviewers on our behalf for their time and effort. We also thank you for your work as the editor. We have also acknowledged editors and anonymous reviewers in the acknowledgments section of the revised manuscript. 

Thanks and regards,

Authors

Response to Reviewers: PONE-D-24-24879

Title: On the hypercomplex numbers and normed division algebras in all dimensions: A unified multiplication

Reviewer #1: 

Comments to authors

The manuscript describes an extension of complex algebra to higher dimensions. Three dimensions are treated in detail. The proposal is very interesting and worth pursuing.

Response: Authors would like to thank the reviewer for providing detailed comments in the previous review round that helped us improve the quality of the manuscript. 

Minor corrections:

p.5 The spherical coordinate system is formally defined in terms of the trigonometric functions, in particular, $\\phi=\\arctan(\\frac{b}{a})$. It is only in computer science where `$\\mathrm{tan2}$' is used. It would be better to use the formal definition and, if required, introduce the computational definition thereafter.

Response: As per suggestion of the reviewer, we have used $\\phi=\\arctan(\\frac{b}{a})$, and introduced the computational definition thereafter.

p.5 Eq. (7) does not follow from Eq. (1)-(2). The wording before Eq. (7) has to be careful to avoid misunderstandings.

Response: As per suggestion of the reviewer, we have changed text before (7) to avoid any misunderstandings.

p.7 Addition and multiplication are the two fundamental operations. Subtraction and division come from inverse elements. Thus, it is customary to define only two operations.

Response: As per suggestion of the reviewer, we have defined addition and multiplication as two fundamental operations, and mentioned that subtraction and division come from inverse elements.

p.8 Figure 2 can be greatly improved. The trajectory is not clear in (b) and (c), where it begins/ends and which path it follows. In fact, maybe (c) could be deferred mentioning that the ZY plane and ZX planes exhibit similar behaviour for the appropriate values of $\\phi$.

Response: Based on the suggestion of the reviewer, we have revised Figure 2, specifically parts (b) and (c). In these updated figures, the left semicircle is depicted in blue, and the right semicircle is in red to illustrate the trajectory. This modification highlights the path traced by $\\exp(j\\theta)$, which follows a semicircular trajectory, as indicated by the blue and red segments for the appropriate values of $\\phi$.

p.9 Eq. (25) $^*$This is the most important revision that is needed$^*$ (Later results will have to be modified accordingly). Let me explain:

Equations (4) to (6) have been carefully devised so that the $\\theta$ domain is $(-\\infty,\\infty)$. Then Eq. (7) is the crucial coordinate definition. The $(-\\infty,\\infty)$ $\\theta$ domain is needed so that the product is a closed operation. However, in (25) what happens if $\\theta$ is larger than $|\\frac{\\pi}{2}|$?

$\\exp(i\\phi)$ and $\\exp(j\\theta)$ are different functions, the ``Euler identity'' type expression for the latter involve your $\\mathrm{cos\\pi}(\\theta)$ and $\\mathrm{sin\\pi}(\\theta)$ functions.

Response: Following the suggestion of reviewer, we have revised the text accordingly. In Equations (7) and (25), in general, \\(\\phi \\in \\mathbb{R}\\), \\(\\theta \\in \\mathbb{R}\\), \\(e^{i\\phi}\\) is \\(2\\pi\\)-periodic and \\(e^{j\\theta}\\) is \\(\\pi\\)-periodic. To normalize these angles, the \\(2\\pi\\) modulo operation is applied to \\(\\phi\\) to restrict it to the principal range \\([0, 2\\pi)\\). Similarly, the \\(\\pi\\) modulo operation is used to adjust \\(\\theta\\) to lie within the principal range \\([- \\pi/2, \\pi/2]\\). Specifically, if \\(\\theta \\notin [-\\pi/2, \\pi/2]\\), it is normalized to this range by adding or subtracting a multiple of \\(\\pi\\), i.e., by applying the transformation \\(\\theta \\mapsto \\theta + n\\pi\\), where \\(n \\in \\mathbb{Z}\\). Therefore, the azimuth angle $\\phi$ is considered $2\\pi$-periodic, and the elevation angle $\\theta$ is considered $\\pi$-periodic as shown in Fig.~4.

p.9 Reconsider your assertion that ``This multiplication is continuous, meaning there are no discontinuities or jumps in the resulting sphere.'' Evaluate, for example the product $(e^{i\\phi_1}e^{j\\theta_1})\\times (e^{j\\theta_2})$, where $\\theta_1=0.9\\frac{\\pi}{2}$ and $\\theta_2=0.2\\frac{\\pi}{2}$. Also, reconsider Remark 2.

Response: Based on the suggestion of the reviewer, we have removed the statement. We have also revised Remark 2 accordingly.

p.9 In line Eq. after Eq. (26) implies $\\phi=0 \\mod \\pi$.

Response: Thank you for suggestion, we have noted it.

p.12 Remark 4 seems unnecessary and it has major problems. Notice that neutral element in the two operations is different.

Response: Thank you for suggestion, we have removed it.

p.12 Eq. (34), has to be revised. Remember that the series representation and the Euler identity are two different things. Since the domain of $\\mathrm{cos\\pi}(\\theta)$ and $\\mathrm{sin\\pi}(\\theta)$ is $(-\\infty,\\infty)$, problems arise since $e^{j\\theta}$ domain has been restricted to $\\pm \\frac{\\pi}{2}$ interval. But if the product is closed it has to be defined in $(-\\infty,\\infty)$, just a $\\mathrm{cos\\pi}(\\theta)$ and $\\mathrm{sin\\pi}(\\theta)$ are. It seems to me that you have to define the $e^{j\\theta}$ series using (4) to (6).

Strictly speaking you should then label this `exponential' function differently, since it differs from the usual complex exponential.

Response: Thank you for suggestion, now we have revised domain of $\\theta$ as $\\theta\\in\\mathbb{R}$ in (34). Moreover, in general, $\\phi \\in \\mathbb{R}$, $\\theta \\in \\mathbb{R}$, $e^{i\\phi}$ is $2\\pi$ periodic and $e^{j\\theta}$ is $\\pi$ periodic. Thus to normalize these angles, the $2\\pi$ modulo operation is applied to $\\phi$ to confine it within the principal range $[0, 2\\pi)$, and the $\\pi$ modulo operation is used to adjust $\\theta$ to lie within the principal range $[-\\pi/2, \\pi/2]$. This suggests that we can also consider $\\theta\\in[-\\pi/2,\\pi/2]$ in (34).

p.14 Some of your assertions and/or proofs have to be carefully worked out. Take for example Proposition1, 8), let $g_1=e^{i\\frac{\\pi}{4}}e^{j\\frac{\\pi}{4}}\\ne 0$ and $g_2=e^{i\\frac{\\pi}{4}}e^{j\\frac{\\pi}{4}}\\ne 0$, what is the value of $g_1g_2=e^{i\\frac{\\pi}{2}}e^{j\\frac{\\pi}{2}}$?

Response: There is a persistent nonuniqueness in representations that arises in specific scenario within the spherical coordinate system (SCS): $Z$-axis $(\\theta = \\pm \\pi/2)$: In this specific scenario where $\\theta$ equals $\\pi/2$ or $-\\pi/2$, for a fixed $r$, the azimuthal angle $\\phi$ can assume any value, as explicitly revealed by equation (2). Consequently, a vector situated along the $Z$-axis possesses an infinite array of representations as $(r, \\phi, \\pm \\pi/2)$. This implies that vectors characterized by diverse values of $r$, $\\phi$, and $\\theta$ can be aligned along the $Z$-axis by adjusting $\\theta$ to $\\pm \\pi/2$. Once aligned along the $Z$-axis, alterations in the polar angle $\\phi$ do not induce any alteration in the orientation of the vector. Thus, in these instances, manipulation of the values of one or more of the other coordinates can be undertaken without inducing displacement of the point in question.

Therefore, if $g_1=e^{i\\frac{\\pi}{4}}e^{j\\frac{\\pi}{4}}\\ne 0$ and $g_2=e^{i\\frac{\\pi}{4}}e^{j\\frac{\\pi}{4}}\\ne 0$, then $g_1g_2=e^{i\\frac{\\pi}{2}}e^{j\\frac{\\pi}{2}}=j$. This is also same as $e^{i\\phi}e^{j\\frac{\\pi}{2}}=j$ for any $\\phi\\in[0,2\\pi)$, which is explained in Eqns. (27), (28) and (29). Moreover, the azimuth angle \\(\\phi\\) in the expression \\(e^{i\\phi}e^{j\\frac{\\pi}{2}}=j\\) indicates that any subsequent variation in the elevation angle \\(\\theta\\) results in the displacement of the point from the \\(Z\\)-axis towards the direction specified by \\(\\phi\\).

Reviewer #2: 

Reviewer #2: In this paper the authors present a comprehensive generalization of complex numbers and the Euler identity in higher dimensions, shedding light on the geometric properties of vectors within these extended spaces. Moreover, they elucidated the practical applications of the proposed methodology as a viable alternative for expressing a quantum state through the multiplication of specified quantum states, thereby offering a potential complement to the established superposition paradigm. Importantly, they explored its utility in point cloud image processing. The paper is well supported by the examples and figures which puts life to the paper. The paper is well written and fits to the scope of the journal. I recommend acceptance.

Response: Authors would like to thank the reviewer for recommendation to accept the paper, and devoting time to provide detailed comments and observations. 

Thanks and Regards,

Pushpendra Singh

(On behalf of all authors)

---

## [Decision Letter · Decision Letter 1]

22 Sep 2024

PONE-D-24-24879R1On the hypercomplex numbers and normed division algebras in all dimensions: A unified multiplicationPLOS ONE

Dear Dr. Singh,

Thank you for submitting your manuscript to PLOS ONE. After careful consideration,  we invite you to submit a revised version of the manuscript that addresses the points raised during the review process.

We look forward to receiving your revised manuscript.

Kind regards,

Sania Asif

Guest Editor

PLOS ONE

Journal Requirements:

Additional Editor Comments:

Dear Prof. Singh,

Based on reviewer #1's report, I suggest authors to go through another round of minor revision and address the attached reviewer's remarks.

Thank you

Best Regards

Sania Asif

Reviewers' comments:

Reviewer's Responses to Questions

**Comments to the Author**

1. If the authors have adequately addressed your comments raised in a previous round of review and you feel that this manuscript is now acceptable for publication, you may indicate that here to bypass the “Comments to the Author” section, enter your conflict of interest statement in the “Confidential to Editor” section, and submit your "Accept" recommendation.

Reviewer #1: All comments have been addressed

2. Is the manuscript technically sound, and do the data support the conclusions?

Reviewer #1: Yes

3. Has the statistical analysis been performed appropriately and rigorously? 

Reviewer #1: Yes

4. Have the authors made all data underlying the findings in their manuscript fully available?

Reviewer #1: Yes

5. Is the manuscript presented in an intelligible fashion and written in standard English?

Reviewer #1: Yes

6. Review Comments to the Author

Reviewer #1: Please send attached comments to authors. A suggestion about how to deal with an exponential function appropriate for their purposes is outlined.

7. PLOS authors have the option to publish the peer review history of their article (what does this mean?). If published, this will include your full peer review and any attached files.

Reviewer #1: **Yes**

---

## [Author Response · Author response to Decision Letter 1]

4 Oct 2024

Response Letter

PLOS ONE

Dear Editor:

First, please accept our thanks for devoting time and coordinating the reviews of the submitted manuscript Ref. No. PONE-D-24-24879 titled ``On the hypercomplex numbers and normed division algebras in all dimensions: A unified multiplication" for possible publication in PLOS ONE. We submitted the manuscript on June 19, 2024, and received review comments from the reviewers on August 31, 2024. The second round of review comments was received on September 23, 2024. We sincerely appreciate the time and effort by the anonymous reviewers dedicated to evaluating our manuscript and offering valuable feedback, which has significantly contributed to enhancing its quality.

We have done our best to address the reviewers' comments. We, hereby, submit: (1) a response letter with detailed replies to each of the reviewers' comments and (2) a revised manuscript that we believe to have significantly improved after incorporating the reviewers' suggestions. 

Please do thank the reviewers on our behalf for their time and effort. We also thank you for your work as the editor. We have also acknowledged editors and anonymous reviewers in the acknowledgments section of the revised manuscript. 

Thanks and regards,

Authors

Response to Reviewers: PONE-D-24-24879}

Title: On the hypercomplex numbers and normed division algebras in all dimensions: A unified multiplication

Reviewer #1

Response: The authors thank the reviewer for providing detailed comments, pointing out errors, and helping us to improve the quality of the manuscript. The authors thank the reviewer for doing some of our homework.

Based on the suggestions of the reviewers, we have incorporated all required changes into the revised manuscript. We have redefined the Equations (4), (5) and (6) as

\\begin{align}

\\tag{4}

 & \\mathrm{\\,cos\\pi}(\\theta)=\\begin{cases}

\\cos(\\theta), & \\text{if } \\theta\\in \\bigcup\\limits_{n=-\\infty}^{\\infty}[-\\frac{\\pi}{2}+2\\pi n,\\frac{\\pi}{2}+2\\pi n],\\\\

-\\cos(\\theta), & \\text{if } \\theta\\in \\bigcup\\limits_{n=-\\infty}^{\\infty}(\\frac{\\pi}{2}+2\\pi n,\\frac{3\\pi}{2}+2\\pi n),

\\end{cases} \\label{cospi1}\\\\

\\text{and } & \\mathrm{\\,sin\\pi}(\\theta)=\\begin{cases}

\\sin(\\theta), & \\text{if } \\theta\\in \\bigcup\\limits_{n=-\\infty}^{\\infty}[-\\frac{\\pi}{2}+2\\pi n,\\frac{\\pi}{2}+2\\pi n],\\\\

-\\sin(\\theta), & \\text{if } \\theta\\in \\bigcup\\limits_{n=-\\infty}^{\\infty}(\\frac{\\pi}{2}+2\\pi n,\\frac{3\\pi}{2}+2\\pi n),

\\end{cases} \\tag{5} \\label{sinpi1}

\\end{align}

where the intervals form a complete, non-overlapping partition of the real line as

\\begin{align}

\\tag{6}

\\mathbb{R} = \\left\\{\\bigcup\\limits_{n=-\\infty}^{\\infty} \\left[ -\\frac{\\pi}{2} + 2n\\pi, \\frac{\\pi}{2} + 2n\\pi \\right]\\right\\} \\bigcup \\left\\{\\bigcup\\limits_{n=-\\infty}^{\\infty} \\left( \\frac{\\pi}{2} + 2n\\pi, \\frac{3\\pi}{2} + 2n\\pi \\right)\\right\\}.

\\end{align}

 %item end 

For the elevation angles, we employ the transformation for all $\\theta\\in\\mathbb{R}$ as follows:

\\begin{align}

\\tag{26}

\\theta \\mapsto \\theta\\, \\mathrm{m\\hat{o}d}\\, \\pi=

\\begin{cases}

\\theta & \\text{if } \\theta\\in [-\\pi/2,\\pi/2],\\\\

\\theta - m\\pi & \\text{if } \\theta > \\frac{\\pi}{2}, \\\\

\\theta + m\\pi & \\text{if } \\theta < -\\frac{\\pi}{2},

\\end{cases} \\label{modpi} 

\\end{align}

where \\( m \\) is the smallest positive integer such that the result lies within the interval \\( \\left[ -\\frac{\\pi}{2}, \\frac{\\pi}{2} \\right] \\).

The standard $\\pi$ modulo function, $\\theta\\mod \\pi$, produces results in the interval $[0, \\pi)$ that can be further adjusted to $[-\\pi/2, \\pi/2)$ or $(-\\pi/2, \\pi/2]$. Thus, $\\theta\\, \\mathrm{m\\hat{o}d}\\, \\pi= \\theta \\mod \\pi$, almost everywhere. Based on the suggestions of the reviewer and above equations, we have revised Result 1 to enhance clarity and eliminate any potential contradictions or confusion. 

We have also added the following relations after Equation (36)

\\begin{align}

 & \\mathrm{\\,cos\\pi}(\\theta)=\\begin{cases}

\\mathrm{Re}\\left(e^{j\\theta}\\right)=\\sum\\limits_{n=0}^\\infty(-1)^{n} \\frac{(\\theta)^{2n}}{(2n)!}, & \\text{if } \\theta\\in \\bigcup\\limits_{n=-\\infty}^{\\infty}[-\\frac{\\pi}{2}+2\\pi n,\\frac{\\pi}{2}+2\\pi n],\\\\

\\mathrm{Re}\\left(-e^{j\\theta}\\right)=-\\sum\\limits_{n=0}^\\infty(-1)^{n} \\frac{(\\theta)^{2n}}{(2n)!}, & \\text{if } \\theta\\in \\bigcup\\limits_{n=-\\infty}^{\\infty}(\\frac{\\pi}{2}+2\\pi n,\\frac{3\\pi}{2}+2\\pi n),

\\end{cases} \\nonumber\\\\

\\text{and } & \\mathrm{\\,sin\\pi}(\\theta)=\\begin{cases}

\\mathrm{Im}\\left(e^{j\\theta}\\right)=\\sum\\limits_{n=0}^\\infty(-1)^{n} \\frac{(\\theta)^{2n+1}}{(2n+1)!}, & \\text{if } \\theta\\in \\bigcup\\limits_{n=-\\infty}^{\\infty}[-\\frac{\\pi}{2}+2\\pi n,\\frac{\\pi}{2}+2\\pi n],\\\\

\\mathrm{Im}\\left(-e^{j\\theta}\\right)=-\\sum\\limits_{n=0}^\\infty(-1)^{n} \\frac{(\\theta)^{2n+1}}{(2n+1)!}, & \\text{if } \\theta\\in \\bigcup\\limits_{n=-\\infty}^{\\infty}(\\frac{\\pi}{2}+2\\pi n,\\frac{3\\pi}{2}+2\\pi n).

\\end{cases} \\nonumber

\\end{align}

Thanks and Regards,

Pushpendra Singh

(On behalf of all authors)

---

## [Decision Letter · Decision Letter 2]

8 Oct 2024

On the hypercomplex numbers and normed division algebras in all dimensions: A unified multiplication

PONE-D-24-24879R2

Dear Dr. Singh,

We’re pleased to inform you that your manuscript has been judged scientifically suitable for publication and will be formally accepted for publication once it meets all outstanding technical requirements.

Kind regards,

Sania Asif

Guest Editor

PLOS ONE

Additional Editor Comments (optional):

Dear Professor Singh,

It's my pleasure to inform you that your paper entitled "On the hypercomplex numbers and normed division algebras in all dimensions: A unified multiplication" (PONE-D-24-24879R2) has been accepted by the editorial board of PLOS ONE. Thanks for your contribution to the Journal.

Best Regards,

Sania Asif

Reviewers' comments:

Reviewer's Responses to Questions

**Comments to the Author**

Reviewer #1: All comments have been addressed

2. Is the manuscript technically sound, and do the data support the conclusions?

Reviewer #1: (No Response)

3. Has the statistical analysis been performed appropriately and rigorously? 

Reviewer #1: (No Response)

4. Have the authors made all data underlying the findings in their manuscript fully available?

Reviewer #1: (No Response)

5. Is the manuscript presented in an intelligible fashion and written in standard English?

Reviewer #1: (No Response)

6. Review Comments to the Author

Reviewer #1: (No Response)

7. PLOS authors have the option to publish the peer review history of their article (what does this mean?). If published, this will include your full peer review and any attached files.

Reviewer #1: No

---

## [Editor Report · Acceptance letter]

14 Oct 2024

PONE-D-24-24879R2 

PLOS ONE

Dear Dr. Singh, 

I'm pleased to inform you that your manuscript has been deemed suitable for publication in PLOS ONE. Congratulations! Your manuscript is now being handed over to our production team.

Kind regards, 

on behalf of

Dr. Sania Asif 

Guest Editor

PLOS ONE